



# Characterization of individual ice nuclei by the single droplet freezing method: a case study in the Asian dust outflow region

Ayumi Iwata[1] and Atsushi Matsuki[2]

[1]Graduate School of Natural Science and Technology, Kanazawa University, Kakuma, Kanazawa, Ishikawa, 920-1192, Japan
[2]Institute of Nature and Environmental Technology, Kanazawa University, Kakuma, Kanazawa, Ishikawa, 920-1192, Japan.

*Correspondence to*: Ayumi Iwata (iayumi@stu.kanazawa-u.ac.jp) and Atsushi Matsuki (matsuki@staff.kanazawa-u.ac.jp)

**Abstract.** In order to better characterize ice-nucleating (IN) aerosol particles in the atmosphere, we investigated the chemical composition, mixing state, and morphology of atmospheric aerosols that nucleate ice under conditions relevant for mixed phase clouds. Five standard mineral dust samples (quartz, K-feldspar, Na-feldspar, Arizona test dust, and Asian dust source particles) were compared with actual aerosol particles collected from the west coast of Japan (Kanazawa City) during Asian dust events in February and April 2016. Following droplet activation by particles deposited on a hydrophobic Si wafer substrate under supersaturated air, individual IN particles were located using an optical microscope by gradually cooling the temperature to -30 ˚C. For the aerosol samples, both the IN active particles and non-active particles were analyzed individually by Atomic Force Microscopy (AFM), micro-Raman spectroscopy, and Scanning Electron Microscopy (SEM) coupled with Energy Dispersive X-ray spectroscopy (EDX). Heterogeneous ice nucleation in all standard mineral dust samples tested in this study was observed at consistently higher temperatures (-25.7 ˚C) than the homogeneous freezing temperature (-36.5 ˚C). Meanwhile, most of the IN active atmospheric particles formed ice below -28 ˚C and were found to be IN active, but slower than the standard mineral dust samples of pure components. The most abundant IN active particles above -30 ˚C were predominantly irregular solid particles that showed clay mineral characteristics (or mixtures of several mineral components). Other than clay, Ca-rich particles internally mixed with other components, such as sulfate, were also regarded as IN active particle types. Moreover, sea salt particles were predominantly found in the non-active fraction, and internal mixing with sea salt clearly acted as a significant inhibiting agent for the ice nucleation activity of mineral dust particles. Also, relatively pure or fresh calcite, $Ca(NO_3)_2$, and $(NH_4)_2SO_4$ particles were more often found in the non-active fraction. In this study, we demonstrated the capability of the combined single droplet freezing method and thorough individual particle analysis to characterize the ice nucleation activity of atmospheric aerosols. We also found that dramatic changes in the particle mixing states during long-range transport had a complex effect on the ice nucleation activity of the host aerosol particles. A case study in the Asian dust outflow region highlighted the need to consider particle mixing states, which can dramatically influence ice nucleation activity.



## 1 Introduction

Ice nucleation in clouds substantially affects the climate by significantly impacting the radiation balance and precipitation processes in the Earth's atmosphere (Lohmann and Feichter, 2005; Rosenfeld et al., 2008; Flato et al., 2013). Most of the initial precipitation process in mid-latitude regions involves ice nucleation in mixed phase clouds, where supercooled water

droplets and ice crystals co-exist (Pruppacher and Klett, 1997; Murray et al., 2012). Therefore, an understanding of ice nucleation is crucial to predicting precipitation and cloud radiative properties.

Pure water droplets generally maintain their liquid state, even in temperatures below 0 ˚C, and remain as supercooled water droplets. These pure water droplets spontaneously freeze by cooling to approximately -37 ˚C and below (homogeneous nucleation) (Murray et al., 2012). However, supercooled water droplets in the atmosphere generally form ice crystals at

higher temperatures due to the presence of aerosol particles that can nucleate ice (heterogeneous nucleation). Traditionally, heterogeneous nucleation pathways are categorized into four freezing modes: deposition, condensation, immersion, and contact freezing modes (Pruppacher and Klett, 1997; Cantrell and Heymsfield, 2005). The physical and chemical properties of aerosols, which act as ice nuclei, play an essential role in the formation of ice crystals. However, the response of ice nucleation processes to changes in host aerosol properties is still poorly understood due to a lack of understanding of the

basic aerosol particle interactions leading to ice crystal formation. Therefore, considerable uncertainty still exists regarding the prediction of ice nucleation that would lead to climate changes in the atmosphere.

Many previous ice nucleation experiments have been performed under laboratory conditions, providing valuable information on the ice nucleation properties of pure component particles and artificially generated aerosol mixtures (Pruppacher and Klett, 1997; Hoose and Möhler, 2012; Murray et al., 2012). Based on these results, mineral dust and biological particles are

generally regarded as efficient ice nuclei (Morris et al., 2004; Connolly et al., 2009; Niemand et al., 2012), while ice nucleation involving soot and organic particles as part of mixed cloud formation is still not clearly indicate because of depending on their composition of particles and experimental conditions (DeMott, 1990; Kireeva et al., 2009). On the other hand, sea salt and sulfates are often not considered as efficient ice nuclei. However, the situation is even more complex in the ambient atmosphere, where particles are often present as complex mixtures of different compounds and minerals.

Although recent laboratory experiments have considered the influence of aerosol mixing states (Sullivan et al., 2010; Kulkarni et al., 2014; Augustin-Bauditz et al., 2016), the complexity of the atmosphere has not yet been fully represented.

Over the last decade, several techniques have emerged, such as the counter vertical impactor (CVI), which is capable of distinguishing ice residue particles from background atmospheric particles (non-active particles) (Klein et al., 2010, Cziczo et al., 2013). Using these techniques, several field studies have been conducted to directly extract ice crystal residues from

cirrus clouds (i.e. pure ice phase). This has enabled the detailed investigation of particles representing the actual deposition mode of ice nuclei in cirrus clouds.

The investigation of ice nuclei in mixed phase clouds (i.e. dominated by immersion and condensation freezing modes), however, requires a different and much more demanding approach to ensure the complete isolation of ice crystals from





numerous supercooled droplets and other interstitial aerosol particles. The accuracy required to ensure complete separation of ice nuclei is very challenging because the number of ice nuclei is believed to be on the order of only 1 out of $10^5$ - $10^6$ ambient particles and supercooled droplets in the free troposphere (Rogers et al., 1998). It is also generally accepted that one single ice crystal grows at the expense of $10^5$ - $10^6$ evaporating droplets inside mixed phase clouds (the Bergeron-Findeisen

or so-called "cold rain" process).

In order to overcome the issue of ice nuclei extraction from mixed phase clouds, several techniques such as the Ice-CVI or the Ice Selective Inlet have been proposed (Mertes et al., 2007; Kupizewski et al., 2014), and efforts have been made to characterize ice nuclei using these instruments (Kamphus et al., 2010; Ebert et al., 2011). Worringen et al. (2015) reported the size distribution and chemical components of ice residue from mixed phase clouds collected by three different techniques

(Ice-CVI, ISI, and ice nuclei counter combined with CVI), and found that silicates, Ca-rich, carbonaceous, and metal oxide particles constitute the major groups of ice nuclei. However, they also reported discrepancies between the results obtained from the three different sampling techniques, and attributed them to potential bias due to the above mentioned artifacts and the scarcity of ice nuclei. Further, it is unknowable whether the ice residues were the actual ice nuclei, because, for example, the riming process can potentially mix cloud condensation nuclei, ice nuclei, and/or interstitial aerosols in the same ice

residue.

In addition to these technical difficulties, the number of field studies that have characterized ice residues in mixed phase cloud remains sparse owing to the limited access to research locations where such clouds are directly and frequently accessible. Therefore, there is a further need to conduct field measurements in many different locations in order to reflect the regional variations in IN particles.

Internal mixing of aerosols commonly takes place during long-range transport in the atmosphere (Zhang et al., 2003). Through internal mixing, the reaction processes, coatings, and aging states at the surface of the particles can dramatically change from their original properties (Trochkine et al., 2003). It is often the case that internal mixing states can vary from particle to particle; therefore, individual particle analysis is necessary for establishing a complete understanding of ice nucleation by ambient aerosol particles.

This study is designed to investigate how the morphology, chemical composition, and mixing state of ambient aerosol particles influence their ice nucleation activities under conditions relevant to mixed phase clouds. Following the ice nucleation experiments, comprehensive analysis of the chemical, physical, and mixing properties of individual aerosol particles collected from the atmosphere was conducted. We further demonstrate the ability to monitor individual IN particles by continuously controlling the ambient conditions during the ice nucleation experiments.

The individual droplet freezing method (IDFM) is the experimental method used in this study, with which ice crystal formation on each particle could be monitored under controlled conditions while keeping individual particles distinct. By drying and evaporating the particles that formed ice crystals and/or droplets, their exact location as ice and/or droplet residues can be observed. Instead of deploying state-of-the-art, in-situ ice nuclei samplers into extreme field locations (e.g.





low temperature, high altitude, and airborne), this method enables detailed post sampling analysis on both IN active and non-active particles on an individual particle basis using a fairly simple, conventional sampling method.

Similar methods have been employed in some laboratory studies to test the ice nucleation activities of various atmospherically relevant standard particles, but for different freezing modes (Fornea et al., 2009; Baustian et al., 2010;

Mason et al., 2015; Whale et al., 2015). In particular, Baustian et al. (2012) investigated individual particles that nucleated ice crystals in the deposition freezing mode by locating the samples in saturated conditions with respect to ice at temperatures between -43 ˚C and -63 ˚C. By employing similar methods, we exposed sample particles to the conditions relevant for mixed phase cloud formation (i.e. immersion and condensation freezing modes), while monitoring the states of individual particles. The sample particles were collected from the west coast of mainland Japan in spring, which is

frequently subjected to the influence of continental outflow; often associated with plumes of Asian dust mixed with bioaerosols and other anthropogenic pollutants (Matsuki et al., 2005; Maki et al., 2010; Tobo et al., 2010).

## 2 Method

### 2.1 Individual droplet freezing method (IDFM)

The sample particles were deposited onto a Si wafer substrate with a hydrophobic coating (Glaco, Soft99 Corporation,

Japan). Particles were observed for their position, size, and shape under an optical microscope with x50 magnification (Olympus, Japan) as shown in Fig. 1a. Subsequently, the substrate was transferred onto a cold stage in a closed cell (Akizawa et al., 2016; THMSG600, Linkam Scientific Instruments, UK).

The dew point of the air introduced into the cell was controlled by mixing dry air (QD10-50, IAC, Japan) and wet air from a water filled gas scrubbing bottle. Both air flows were kept particle free and their mixing ratio was controlled by a pair of

mass flow controllers (MQV9005, Azbil Corporation, Japan). The resulting dew point was also monitored by a chilled mirror type hygrometer (OptiSonde[TM], General Electric Company, Japan). By adjusting both the dew point of the introduced air flow (0.5 l/min) in the range -6 ˚C to -3 ˚C and the sample temperature on the cold stage in the range -9 ˚C to -7 ˚C, the particles on the substrate were exposed to water super saturation conditions that initiate droplet formation. Thus, increases in particle sizes could be observed (Fig. 1b).

After the air supply was stopped, the temperature of the stage was reduced to -30 ˚C at a rate of -0.5 ˚C/s. As the temperature of the stage decreased, the saturated water vapor was expected to keep condensing onto the droplets. Thus, the degree of water super saturation was assumed to be limited to slightly higher than 100 % in terms of relative humidity. The formation of ice crystals on individual droplets can be visually identified by their rapidly growing size with irregular shapes (Fig. 1c).

After relaxing for 10 seconds once the substrate reached -30 ˚C, the temperature of the stage was increased up to -10 ˚C at

0.5 ˚C/s. After reaching this temperature, dry air flow (0.5 l/min) was introduced into the cell to expose the formed ice crystals and droplets to the sub-saturation conditions for ice. As a result of evaporation and/or sublimation of water, the nuclei particles were exposed and left visible on the substrate. Finally, the positions of the dried particles were again located



under the optical microscope (Fig. 1d). By comparing the optical images before and after the ice nucleation experiments, the individual particles that formed ice crystals (excluding those coalescing with adjacent droplets or crystals) were identified and regarded as IN active particles.

### 2.2 Preparation of atmospheric aerosol and standard particles

Using the IDFM, we identified IN particles from the ambient aerosol particles and five types of standard samples. We sampled actual aerosol particles at Kanazawa University campus (36.54 ˚N, 136.70 ˚E, 149 m. ASL), Japan (Fig. 2) on 28 February 2016 and 10 April 2016. The particles were collected on the substrate (described above) using an impactor with a 50 % cutoff diameter of 1.1 µm at a flow rate of 1.0 L/min. The sampling period was set to 60 seconds for each substrate. The ambient temperature, pressure, relative humidity, and particle number concentrations were recorded using a mobile

meteorological sensor (TR-72Ui, T&D Corporation, Japan) and an optical particle sizer (OPS 3330, TSI, USA) during the sampling periods.

The ambient atmospheric conditions during the ambient aerosol sampling were 16.7 ˚C, 22.7 % RH, and 1004.25 hPa for the February sample, and 11.7 ˚C, 84.0 %, and 994.47 hPa for the April sample. The average concentrations of coarse particles during the two sampling periods (Dp > 1.117 µm) were approximately 1.4E + 02 particles cm$^{-3}$ for February and 4.1E + 02

particles cm$^{-3}$ for April, indicating that the concentration for April was three times higher than for the February sampling period. No significant difference was observed for the fine particles (Dp < 1.117 µm) between the two periods.

The five standard samples include three types of single mineral component samples (quartz, K-feldspar, Na-feldspar) and two types of soil dust samples (Arizona test dust: ATD, and Asian dust source particles: ADS) that consist of multiple mineral components. The ADS were sampled from the surface soil of an arid region near Dunhung, China (40.21 ˚N, 94.68

˚E). K-feldspar and Na-feldspar were purchased from the Bureau of Analyzed Samples Ltd. K-feldspar has been reported by several studies to show higher ice nucleation activity in mineral dust (Atkinson et al., 2013; Harrison et al., 2016). The quartz sample was purchased from Wako Pure Chemical Industries, Ltd., and it was further crushed with a mortar to decrease the grain size. Both of the soil dust samples (ATD and ADS) were size selected by the impactor and only particles coarser than 1.1 µm were used in the ice nucleation experiments.

### 2.3 Individual particle analyses

Both the atmospheric particles that IN active and non-active particles were analyzed on an individual particle basis using an AFM (CombiScope$^{TM}$ 1000, AIST-NT, Inc., USA) and by micro-Raman spectroscopy (Nanofinder$^®$HE, Tokyo Instruments, Inc., Japan) to characterize the three-dimensional morphology and detect surface chemical compounds, respectively. Furthermore, the exact same particles were analyzed by a SEM (S-3000N, HITACHI, Japan), coupled with EDX (EMAX-

500, HORIBA, Japan), in order to obtain their elemental compositions. The reason for conducting AFM observations and micro-Raman analysis prior to SEM-EDX analysis is because the former two methods can be applied under ambient conditions, while the latter requires a high vacuum. The sequence of the multiple analyses was determined by taking into





account the potential loss of volatile components within individual particles, especially under conditions of high vacuum and electron beam bombardment. Further, the advantages of AFM over conventional imaging techniques are that it is non-destructive (ambient conditions, no electron beam bombardment, and minimum physical contact), and directly provides the three-dimensional structure of individual particles. The technique does not involve any pre-treatment (e.g. shadowing

particles by spattering a Pt/Pd alloy from a certain angle inside the vacuum) or tilting of samples as is conventionally required, especially when obtaining particle height from the inherently two-dimensional electron micrographs (Adachi et al., 2007; Ueda et al., 2011).

In this study, AFM images were obtained in dynamic mode using a silicon tip (ATEC-NC, NANOSENSORS™, Switzerland) with a spring constant of 45 N/m. The resolution of the measurement was set to 20 nm for each particle. The tip

was resonated at approximately 335 kHz. In order to avoid sweeping of the particles by physical contact with the tip, the amplitude of the tip was set to 200 nm, and the rate of scanning speed was set to 0.1 lines per second.

The Raman spectra of individual particles were obtained using a 532 nm excitation laser with the intensity fixed at 4.906 mW. With the use of a 100x objective lens, the laser spot size (i.e. spatial resolution) approaches the diffraction limit of approximately 1 μm in diameter. The laser was scanned over the Si wafer substrate containing the particle deposits with an

automatically controlled X, Y piezo scanner stage. For each 750 nm step, a Raman spectrum was acquired with an exposure time of 10 s and 5 accumulations.

The compounds and minerals contained in the particles were identified by the main Raman-shift peaks in the spectral range - 100 $cm^{-1}$ to 4,000 $cm^{-1}$. The peaks were assigned to specific compounds and minerals by comparing with reported values from the literature (Tang and Fung, 1989; Ivleva et al., 2007; Freeman et al., 2008; Baustian et al., 2012; Laskina et al.,

2013) and Raman databases. The peak positions were also verified by measuring standard pure chemical compounds using the current Raman system. The Raman-shift peak positions used for the identification of compounds and minerals are summarized in Table 1. The identification of organic components was defined by detection of the C-H vibrational mode that appears as peaks or a broad peak in the range 2,800 $cm^{-1}$ and 3,100 $cm^{-1}$ (Baustian et al., 2012; Ault et al., 2013; Laskina et al., 2013). The strong broad peak obtained from atmospheric particles in the range 1,200 $cm^{-1}$ and 1,700 $cm^{-1}$, such as that

shown in Fig. 3a and c, is typically due to the presence of complex organic matter with conjugate double bounds, biological material, diesel soot, black and brown carbon, or humic-like substances (Escribano et al., 2001; Sadezky et al., 2005; Ivleva et al., 2007). Therefore, we refer to this broad peak as black or brown carbon (BBC) (Hiranuma et al., 2011). Other than peaks assigned to the specific components mentioned here, spectra showing strong fluorescence in the measured range, as shown in Fig. 3a, were further classified as "fluorescent particles".

Following the measurements under atmospheric pressure by AFM and micro-Raman, the particles on the Si wafer substrate were coated by Au approximately 30 nm thick. The coated particles were located under SEM-EDX. The X-ray spectra were collected at 20 kV acceleration voltage and 15 mm working distance. The relative atomic fractions (%) of the detected elements were determined by the ratios of the characteristic X-ray peak areas. Due to the limitations of the method, related to



the quantification of lighter elements and elements contained in the coating and substrate, C, N, O, Au, and Si were excluded from the semi-quantitative analysis.

## 3 Results

### 3.1 Observation of ice nucleation by IDFM

Before evaluating the ice nucleation activity of the standard samples by IDFM, we measured the freezing temperature of pure water droplets using the same method, which can be regarded as the onset temperature of homogeneous freezing. As a result, homogeneous ice nucleation was initiated at approximately -36.5 ˚C, and all the pure water droplets were frozen by -40 ˚C. This homogeneous freezing temperature coincides with those reported by several previous laboratory experiments (Pruppacher and Klett, 1997; Murray et al., 2010; Murray et al., 2012) and from observations of deep convective clouds
(Rosenfeld and Woodley, 2000).

Heterogeneous ice nucleation observed in all standard samples tested in this study consistently occurred at higher temperatures than the homogeneous freezing temperature. The standard samples of single mineral components included K-feldspar, Na-feldspar, and quartz, and their freezing onset temperatures were -20.3 ˚C, -20.7 ˚C, and -25.7 ˚C, respectively. Therefore, the ice nucleation activity of K-feldspar was the highest and that of quartz was the lowest. The order and the
range of observed onset temperatures for these minerals were consistent with the freezing temperatures based on micro-liter suspension droplets by Atkinson et al. (2013). The fact that the observed freezing temperatures were similar to those reported in previous studies clearly demonstrates the validity of the IDFM for representing immersion and condensation mode ice nucleation. For comparison, the freezing temperatures of ATD and ADS were -22.5 ˚C and -26.6 ˚C, respectively.

For ambient samples, we first determined the total number of target particles by analyzing the recorded optical images. As a
result, ice nucleation activity of 10,188 and 24,145 particles were monitored by the IDFM for the February and April samples, respectively. Among those, the number of IN active particles was 37 for the February samples and 122 for the April samples. Therefore, the ice nucleation active fraction for $D_p > 1.1$ μm particles was 5.6E -03 for the April samples. This value is 1.5 times higher than that for the February samples (3.6E -03). While most of the IN active particles formed ice below -28 ˚C, the highest onset temperature recorded for these ambient particles was -25 ˚C. Particles with such a relatively
high onset temperature tended to grow faster and larger, therefore coalescing with adjacent particles, complicating identification of individual particles following the ice nucleation experiment. As a result, individual analysis of such rapidly growing particles was not performed.

### 3.2 Observation of individual IN active particles by AFM

Owing to the inherently small number of IN particles in the atmosphere, three different analytical methods were employed to
gather as many physical and chemical characteristics as possible from individual IN active particles.





Firstly, the 3D morphological images and the maximum height (h) based on the cross-sectional shape of 22 IN active particles and 67 non-active particles were obtained by AFM (note that the counts are shown as the sum of February and April samples due to the small number of IN active particles). We further determined the surface equivalent diameter ($D_S$), which is defined as the arithmetic average of the longest axis and its orthogonal axis in the 2D particle silhouette. As a result,

the IN active particles were found to fall within the 1.12 - 14.60 µm range for $D_S$ and in the 0.32 - 3.28 µm range for h, while the non-active particles had $D_S$ values of 0.60 - 12.04 µm and h values of 0.07 - 1.98 µm.

In terms of particle shape, we determined the $D_S$/h ratio, which can be closely linked to the physical state, whereby semi-ellipsoidal particles with high $D_S$/h ratios tend to be in a liquid or semi-liquid state, while irregular shaped particles with low ratios are most likely in a dry solid phase (Sobanska et al., 2014). As shown in Fig. 4a, while the IN active particles tended to

concentrate in low $D_S$/h ratios, the non-active particles showed a wider range, including relatively high $D_S$/h ratios. This suggests that the IN active particles were predominantly irregular solid particles.

Aspect ratio was also determined for each particle using the ratio between the major and minor axis of the best fit ellipse of the 2D particle silhouette. Although the number of measured particles was small, the aspect ratio of the IN active particles was not close to 1 (Fig. 4b). In other words, the IN active particles were predominantly irregularly shaped particles (as

shown in Fig. 3a), consistent with the results of $D_S$/h ratio distribution (Fig. 4a). Conversely, the particles shown in Fig. 3b and c were more representative of non-active particles. It is noteworthy, however, that the analyzed morphology of particles may not necessarily correspond to that in the actual atmosphere, since particles can be flattened upon impaction and particle shape can also be altered following ice nucleation experiments involving ice and/or droplet activation.

### 3.3 Analysis of individual IN active particles by micro-Raman spectroscopy

Chemical species contained in the 42 IN active particles and 131 non-active particles were identified by Raman spectra mapping (counts include the February and April samples combined). The detection frequencies of the Raman-active molecular compounds among individual particles are summarized in Fig. 5.

It was found that more than 70 % of both IN active particles and non-active particles contained organic matter. Additionally, nitrate and/or sulfate peaks were detected in 50 % and 99 % of the IN active and non-active particles, respectively. These

results show that most of the ambient aerosol particles ($D_P$ > 1.1 µm) collected in this study were internally mixed with organic and/or inorganic (sulfates and nitrates) materials.

The fraction of particles containing sulfates, such as $(NH_4)_2SO_4$ and/or $CaSO_4$, was 40 % and 34 % for IN active and non-active particles, respectively. Moreover, a significantly larger fraction (76 %) of the IN active particles showed fluorescence in the Raman spectra. Fluorescence from a particle is typically attributed to a certain group of organics of biological origin,

or the intercalated impurities of humic or humic-like substances in clay minerals and amorphous alumino-silicates (Sobanska et al., 2012; Jung et al., 2014). In addition to the fraction of fluorescent particles, BBC, $CaSO_4$, and quartz were detected in the IN active particles with significantly higher frequencies than in non-active particles. In contrast, the particles indicating





broad OH peaks and nitrate peaks were more abundant in non-active particles. In terms of the detection frequency of organic matter and $(NH_4)_2SO_4$, we did not find any clear differences between IN active and non-active particles.

We also performed micro-Raman analysis on ADS for comparison. As shown in Fig. 5, all ADS showed fluorescence in the Raman spectra. Organic matter and quartz components were identified in 61 % and 16 % of individual ADS, respectively.

Meanwhile, peaks indicating sulfates or nitrates were not detected in any ADS particles. This is consistent with the findings that Asian dust aerosols near the source region are fresher and hence contain less sulfur than in the downwind regions (Trochkine et al., 2003). This result clearly indicates that many mineral dust particles are originally internally mixed with organic matter to some extent, but that the Asian dust source does not contain sulfates or nitrates (Kawamura et al., 2004).

Additionally, the presence of feldspar in the individual particles was tested by comparing spectra with that of the standard

mineral samples. However, there were almost no particles showing typical spectra indicating the presence of feldspar in the ambient samples. The possibility remains that the feldspar content was so small that it was below detection limits, or that the feldspar contained impurities or had defects in the crystal structures that may have caused interference in the form of strong fluorescence in the Raman spectrum. Nonetheless, we did not observe any particles predominantly composed of pure feldspar, such as the standard feldspar sample, that indicated characteristic peaks in the Raman spectra.

### 3.4 Analysis of individual IN active particles by SEM-EDX

Elemental compositions of 37 IN active and 114 non-active particles were analyzed using SEM-EDX (counts include the February and April samples combined). The presence of mineral dust particles can be identified by the dominant X-ray peaks corresponding to Al, Mg, and Fe. Due to interference by the Si wafer substrate, Si was not determined in this study. The particles predominantly composed of Na and Cl were classified as fresh sea salt with a Cl/Na ratio in the 0.8 - 1.2 range.

Due to the Cl liberation reaction during transport (Zhang et al., 2003), those particles with lower Cl/Na ratio can be considered as aged sea salt particles. Particles enriched in Ca or S were classified as Ca-rich particles or sulfate particles, respectively. When a particle contained more than 35 % of elements other than the predominant components, we defined it as an internal mixture, which is designated by (+ inclusion) in Fig. 6.

As shown in Fig. 6, the relative abundance of particle groups clearly differed between IN active and non-active particles.

The mineral dust particle groups (mineral dust and mineral dust + inclusions) accounted for 55 % of IN active particles and were the most dominant type, while sea salt particles were rarely found (3 %). Conversely, the majority (62 %) of non-active particles was dominated by fresh and aged sea salt particles, suggesting it was internally mixed with other components, such as S. The mineral dust particle group, in turn, comprised a relatively minor fraction (12 %).

The Ca-rich and sulfate groups were found in both IN active and non-active particles. However, the pure Ca-rich and sulfate

component groups were relatively minor compared to internal mixtures, which contain the latter components plus other matter (+ inclusion), for IN active particles. They were found in similar fractions for non-active particles.

Particles enriched in Pb were reported to be involved in the formation of ice crystals at Jungfraujoch (Cziczo et al., 2009). However, such particles enriched in Pb were neither found in IN active particles nor in non-active particles.





The semi-quantitative comparison of particle composition obtained by micro-Raman spectroscopy and SEM-EDX analyses is hampered by the different principles employed in the two techniques. Particle classification with SEM-EDX relies on characteristic X-ray signals, which are used to estimate the elemental composition of a particle. This information can be considered to reflect the bulk elemental distribution within a particle and has been commonly used for particle classification

in many previous studies. In contrast, micro-Raman spectroscopy detects slight shifts of wavelength that reflect the vibrations of molecular bonds contained in a sample. While the laser transmission depth depends on the particle material, and it is not necessarily representative of a major component of the particle. In the following discussion, we considered that micro-Raman spectra are typically representative of a coating and/or components concentrated on the particle surface, i.e. the analytical volume is more surficial than for EDX.

**4 Discussion**

**4.1 Mineral dust particles**

The SEM-EDX analysis indicated that mineral dust particles accounted for 55 % of the IN active particles (Fig. 6). Fluorescent particles identified by micro-Raman spectroscopy were also common (76 %) in IN active particles (Fig. 5). Subsequent SEM-EDX analysis showed that most of the fluorescent particles (87 %) contained elements that indicate a

mineral composition. Considering the fact that all the ADS also showed similar fluorescence, these fluorescent particles could be associated with mineral dust (especially those enriched in clay minerals). Both the SEM-EDX and micro-Raman analyses indicated that mineral dust particles act as efficient ice nuclei under conditions relevant for mixed phase cloud formation.

This finding is in good agreement with the findings of previous work. For example, several laboratory studies showed that

mineral dust particles are capable of nucleating ice crystals at relatively high temperatures, and may be important ice nuclei, especially at temperatures below approximately -15 ˚C (Murray et al., 2012). Additionally, several in-situ field studies performed within a mixed phase cloud at Jungfraujoch, a high elevation site in the Swiss alpine region, reported that mineral dust was the most abundant component, comprising 40 - 70 % of the ice residue particles by number (Worringen et al., 2015; Ebert et al., 2011; Kamphus et al., 2010).

Nonetheless, questions remain concerning the factors controlling the IN properties of mineral dust. These particles exist as internal mixtures of several minerals. Strictly speaking, therefore, the mineralogical composition of mineral dust is unique to each particle. Recently, several studies have pointed out that the feldspar group, especially K-feldspar, is an important mineral component for ice nucleation within mixed phase clouds at higher temperatures (Atkinson et al., 2013; Harrison et al., 2016; Boose et al., 2016). Also, quartz has been proposed to contribute to the relatively strong ice nucleation activity of

natural desert dusts (Boose et al., 2016), whose stronger ice nucleation activity in immersion mode than that of clay minerals is thought to be related to defects present on the surface of quartz particles (Zolles et al., 2015).



In this study, standard K-feldspar (-20.3 ˚C) and quartz (-25.7 ˚C) samples were indeed more efficient ice nuclei than ADS (-26.6 ˚C). However, most of the collected atmospheric particles formed ice crystals at temperatures below -25 ˚C. These ice nucleation onset temperatures are similar to those reported for several natural dust samples from different regions around the world (Boose et al., 2016; Kaufmann et al., 2016).

The mineral dust particles can be further separated into K-feldspar and Na/Ca-feldspar groups using an Al-K-(Ca+Na) ternary plot (Fig. 7). Mineral particles composed mainly of other clay minerals and mica should appear near the center of the Al-K-(Ca+Na) ternary plot. However, K-feldspar dominated particles were rarely observed in the atmospheric mineral dust particles analyzed in this study, although Na/Ca-feldspar (plagioclase feldspar), mica, and clay minerals were present. Also, peaks of feldspar (2 %) or quartz (19 %) were rarely identified in the Raman spectra of the analyzed mineral dust particles,

further suggesting the absence of K-feldspar enriched particles. Even where feldspar and quartz peaks were observed, they also contained other mixtures such as fluorescent material, organic matter, or sulfates, which were not particularly IN active. Therefore, highly IN active dust particles composed of single component minerals, such as feldspar or quartz, were very rare or non-existent in the ambient aerosols collected in this study. Although we may have miscounted some IN active particles due to the aforementioned coalescence with adjacent particles during ice crystal growth, even the most rapidly growing

crystals initiated ice nucleation below - 25 ˚C. Thus, it would have been noticed if particles as active as K-feldspar were nucleating at temperatures as high as -20.3 ˚C.

Previous studies have reported that clay and mica account for 51 % of the Asian dust particles over Japan, while quartz (10 %) and feldspar (5 %) are minor mineral components, based on similar identification criteria to that employed in this study (Iwasaka et al., 2009). Simulated Asian mineral dust has also been investigated, and the detection frequency of K-

feldspar was significantly lower (2 %) than that of Na/Ca-feldspar (12 %) (Yabuki et al., 2002). The results of onset temperature and related particle mineralogy (both in terms of SEM-EDX and micro-Raman spectroscopy), suggested that IN mineral dust particles as efficient as pure component K-feldspar or quartz are extremely rare in the atmosphere (at least in East Asia, which is affected by Asian dust). Furthermore, it was demonstrated that most IN particles above -30 ˚C were mineral dust particles composed mainly of clay minerals, with or without minor mixing of other mineral components, that

involve fluorescence, most likely as a result of defects and/or impurities (e.g. humic organics) in their crystal structure (Gaft et al., 2005; Jung et al., 2014; Sovanska et al., 2014).

More recently, Kaufmann et al. (2016) found no significant differences between the freezing temperatures of dust samples collected from ground soil of various arid dust source regions. Mineral components with extremely high ice nucleation activity, such as the alkali feldspar microcline, were found only as minor components. Their results further suggested that

dust mixing in the natural environment reduces high ice nucleation efficiency (Kaufmann et al., 2016). Additionally, minerals in the natural environment are decomposed by reactions with water (chemical weathering processes), thus forming Al-rich clay minerals.  Our results are consistent with these findings, such that the distribution of freezing temperatures of the studied natural dusts is much more compact and falls within a narrower temperature range than that reported for a variety of reference minerals. It is worth noting that our results are based on actual atmospheric aerosol samples rather than sieved





surface soil, and that the freezing experiment and characterization were made on a single particle basis by the coupling of IDFM, micro-Raman spectroscopy, and SEM-EDX. Therefore, our results should be representative of the ice nucleation activity expected within mixed phase clouds.

### 4.2 Sea salt particles

While mineral dust particles were found with the highest frequency among the particles that nucleate ice crystals, sea salt particles were the most dominant particle type in non-active particles (Fig. 6).

Although sea salt particles were assigned based on the largest atomic concentration of Na and Cl by EDX analysis, many particles had a combined Na and Cl concentration of less than 65 % with low Cl/Na ratios. This is an indication that many sea salt particles were aged and mixed with other components. In EDX analysis, in particular, most of these sea salt mixtures

were characterized by a large S fraction, indicating that they were internally mixed with sulfates. Moreover, those particles showed peaks of organic matter, sulfates, or nitrates (in particular $MgNO_3$), in the Raman spectra. Therefore, these particles can be considered as aged sea salt particles that were internally mixed with nitrates, sulfates, or organics. The freezing experiments clearly demonstrated that these aged sea salt particles are not efficient ice nuclei in mixed phase cloud formation.

Ice nucleation by sea salt particles has been suggested by previous field studies but under conditions relevant for cirrus cloud formation instead (Cziczo et al., 2013). Meanwhile, ice nucleation by sea salt particles under mixed phase cloud conditions has not been fully confirmed by previous field studies, due to the geographical limitation of research facilities and technical difficulties related to the direct collection of ice residue particles (Worringen et al., 2015). Laboratory studies have reported ice nucleation by crystalline sea salt at much lower temperatures (Wise et al., 2012). Recently, Wilson et al. (2015) reported

that sea spray organic aerosols derived from the sea surface microlayer nucleate ice under conditions relevant for mixed phase clouds and ice cloud formation at high-altitudes, but the sea water itself did not contribute markedly to ice nucleation. Based on the results of this study, we suggest that large and aged sea salt particles internally mixed with sulfates, nitrates, or organics will not nucleate ice in any event, although the possibility remains for ice nucleation by pure sea salt and sea spray organic particles in the atmosphere.

A ternary diagram of Na-(Al+Mg+Fe)-(Ca+S) showing the compositions of all analyzed particles is shown in Fig. 8. Interestingly, the diagram clearly indicates that particles with Na greater than 35 % were dominant amongst non-active particles. This indicates that internal mixing with sea salt may act as an important inhibiting factor for ice nucleation within mixed phase clouds. For example, particles containing mineral dust components (often found among IN active particles), showing clear mixing with sea salt, were rarely found among IN active particles, but were rather common in the non-active

particle group.





### 4.3 Ca-rich and sulfate particles

As shown in Fig. 6, both relatively pure Ca-rich particles and sulfate particles (without inclusions) were detected with similar frequencies among the IN active and non-active particles. The particles enriched in Ca can be related to mineral dusts such as calcite ($CaCO_3$), dolomite ($CaMg(CO_3)_2$), and gypsum ($CaSO_4$). Calcite or dolomite dominated particles can be

inferred by the elemental ratio of Mg/Ca (calcite: Mg/Ca < 0.5, dolomite: Mg/Ca > 0.5). The presence of pure gypsum can be inferred from the S/Ca ratio of 1 in the Ca-rich particle group (Trochkine et al., 2003), but a S/Ca ratio of < 1 is expected because gypsum is typically regarded as occurring in a mixture with calcite (Iwasaka et al., 2009). The dolomite dominated particles with high Mg contents were rare among the Ca-rich particles. Most of the Ca-rich particles were classified as calcite or gypsum with Mg as a minor component.

There seems to be no marked difference between the IN active and non-active Ca-rich particles in terms of their mixing states with other mineral components (e.g. Al, Mg, Fe) from the SEM-EDX analysis (Fig. 9). Interestingly, however, Ca-rich particles with small S content (S/Ca < 0.2), which can be regarded as predominantly calcite, were more common in non-active particles (i.e. bottom right corner in Fig. 9). Of the non-active Ca-rich particles, 50 % were detected with a carbonate peak in the Raman spectra, confirming the presence of calcite. In contrast, carbonate peaks were hardly detected in Ca-rich

particles in the IN active group. In terms of compounds identified by micro-Raman analysis, the detection frequencies of fluorescence, sulfates, or organic matter among Ca-rich particles were difficult to compare as no obvious differences were found between the IN active and non-active particles. Meanwhile, nitrate peaks, in particular the $Ca(NO_3)_2$ peak, were more frequently detected in non-active particles than in IN active Ca-rich particles. In other words, we suggest that Ca-rich particles mixed with nitrates form ice much less easily than Ca-rich particles mixed with organics and sulfates.

Frequent detection of $Ca(NO_3)_2$ has several important implications related to the physical state of the Ca-rich particles in the atmosphere. It is reported that originally solid Ca-rich ($CaCO_3$) particles readily deliquesce and are converted into aqueous droplets following the atmospheric reaction with gaseous $HNO_3$ to form $Ca(NO_3)_2$ (Laskin et al., 2005; Matsuki et al., 2005). Indeed, we found that the aspect ratios of non-active Ca-rich particles measured by AFM tended to show smaller values closer to 1. Furthermore, AFM topographic images of such particles typically showed thin coatings around the core (core-

shell structure) as shown in Fig. 10b, very much resembling the morphological features of the Ca-rich aqueous droplets described in the literature. Note also that the core of the particle showed a peak for Al in the EDX spectra, suggesting the presence of alumino-silicate (e.g. clay) particle in the center. Such Ca-rich spherical particles were exclusively found among non-active particles rather than the IN active Ca-rich particles (Fig. 10a), suggesting that nitrate containing Ca-rich spherical particles are less likely to act as ice nuclei in the mixed phase clouds. It is worth pointing out that, $Ca(NO_3)_2$ is an extremely

soluble salt with a solubility of 121.2 g/100g $H_2O$ at 20 ˚C. Therefore, the liquefied $Ca(NO_3)_2$ coating is expected to show strong molar depression of freezing point, which could explain their weak IN ability. Therefore, these Ca-rich spherical particles mixed with nitrates were found to be less IN active than Ca-rich particles mixed with organics or sulfates, regardless of the composition of their core particles (e.g. with or without clay mineral particles). Tobo et al. (2010) reported



similar liquefied particles can be formed through reaction with gaseous HCl to form CaCl$_2$. Such chloride particles are also expected to behave similarly to the Ca-rich particles having Ca(NO$_3$)$_2$.

There is still a slight discrepancy among previous studies concerning the ice nucleation activity of carbonates. For example, several laboratory studies regarded calcite as an inefficient ice nuclei (Murray et al., 2012; Atkinson et al., 2013), while one laboratory experiment and field study reported that Ca-rich particles act as ice nuclei (Zimmermann et al., 2008; Worringen et al., 2015). Although the number of analyzed particles is rather small, our result suggests that pure calcite (i.e. fresh) particles or their mixture with nitrate are less likely to nucleate ice under mixed phase cloud conditions.

Particles with S/Ca > 0.8 were grouped into sulfate particles. In terms of elemental composition determined by SEM-EDX analysis, there were no differences between the sulfate dominated particles that were IN active and non-active. However, from the micro-Raman analysis, 43 % of non-active sulfate particles showed a peak of (NH$_4$)$_2$SO$_4$, while none of the IN active sulfate particles showed this peak. Particles identified as CaSO$_4$, or those with fluorescent peaks, showed a relatively higher abundance among IN active particles.

Sulfate particles are generally believed to nucleate ice crystals only under cirrus forming conditions (Abbatt et al., 2006). However, field studies have reported the presence of sulfate particles in ice residues, although potential bias from sampling artifacts cannot be entirely excluded (Prenni et al., 2009; Worringen et al., 2015; Ebert et al. 2011).

With regard to the Ca-rich and sulfate particles studied here, in summary, our results suggest that (NH$_4$)$_2$SO$_4$, intact calcite, or Ca-rich spherical particles mixed with nitrates in the atmosphere are less likely to nucleate ice. Meanwhile, Ca-rich particles with internally mixed sulfate (e.g. CaSO$_4$), or sulfate dominated particles internally mixed with clay minerals, may have a higher chance of nucleating ice under mixed phase cloud formation conditions.

## 4.4 Organic matter

Various organic matter are believed to be involved in ice nucleation. For example, some primary biological particles such as bacteria and pollen can act as efficient ice nuclei (Möhler et al., 2008a; Pummer et al., 2012; Hara et al., 2016). Soot particles and humic-like substances also act as ice nuclei in immersion mode, although with variable freezing temperatures (Diehl and Mitra, 1998; DeMott et al., 1999; Brooks et al., 2014; O'Sullivan et al., 2014). Based on Raman spectroscopy, 70 % of particles showed the broad peak indicating the presence of organic matter in both the IN active and non-active particles analyzed in this study. The frequent detection of organic matter in this study is in agreement with other field measurements on atmospheric particles (Baustian et al., 2012).

Particles detected with and without organic matter showed no significant difference in terms of the particle groups and their fractions identified by SEM-EDX for both IN active and non-active particles. This implies that the organics detected in this study had a secondary effect on ice nucleation above -30 ˚C, or simply that the current definition of organic matter (based on the C-H stretching band alone) is not capable of resolving IN active and non-active compounds.

In the Raman analysis, 98 % of particles detected with a broad peak arising from the intermolecular hydroxyl groups (3,200-3,650 cm$^{-1}$) were associated with particles containing the organic matter peak (C-H). It is worth noting that a similar





association of hydroxyl groups with organic matter was reported for IN particles under cirrus cloud conditions (Baustian et al., 2012). The detected OH broad peaks associated with organic matter are not as strong as the Raman intensities from water typically observed with particle deliquescence and, therefore, clearly distinguishable. As shown in Fig. 5, non-active particles (53 %) showed a higher detection frequency of organic particles with OH broad peaks than IN active particles

5 (39 %).

Past laboratory studies reported that biomass burning particles, organic acid, and secondary organic particles have the potential to nucleate ice in the conditions relevant for cirrus cloud formation (Petters et al., 2009; Prenni et al., 2009b; Hoose and Möhler, 2012). Möhler et al. (2008b) suggested that thick coatings of organic matter can inhibit the depositional freezing of mineral dust. Moreover, the ice nucleation activity of mineral dust particles is enhanced due to mixing with heat sensitive

organic substances supposedly of biological origin (Conen et al., 2011; Tobo et al., 2014). The ice nucleation activity of oxidized organic matter has also been reported from several laboratory experiments and field studies, but for conditions of cirrus cloud formation (Knopf et al., 2010; Baustian et al., 2012). In this respect, the effects of mixing with either oxidized or non-oxidized organic matter on ice nucleation in mixed phase clouds have not been fully confirmed, either by laboratory or field experiments.

Nonetheless, the impact of major particle components other than the organic matter (e.g. mineral dust and sea salt) were more apparent in terms of the different ice nucleation activity observed under simulated mixed phase cloud conditions. Therefore, we suggest that observed ice nucleation activity above -30 ˚C cannot be linked directly to the oxidative state of the organic matter on the surface of the particles. Instead, ice nucleation activity may be controlled more strongly by the major components governing their particle types. Furthermore, the IN active and non-active fractions showed different

frequencies in terms of particles containing oxidized and less oxidized organic matter. Also, IN active particles were more often associated with BBC peaks, which are suggestive of the presence of complex organic matter. The possibility remains, therefore, that less oxidized organic matter and BBC may partly be responsible for, or even enhance, the ice nucleation activity of the host particles.

### 4.4 Sampling conditions: influence of aged Asian dust particles

The study area, Kanazawa City, is located along the west coast of mainland Japan, and further upstream of the westerly continental outflow are the vast arid regions of inland China and Mongolia. Every spring, frequent dust outbreaks are observed, transporting a massive amount of mineral dust aerosols (Asian dust) across the region and beyond (Iwasaka et al., 2009).

Backward trajectories suggested that the air mass at 3,000 m altitude over the sampling location during the two sampling

dates in February and April 2016, had traveled over Chinese and Mongolian regions (Fig. 11; redrawn from http://ready.arl.noaa.gov/HYSPLIT.php; Stein et al., 2015; Rolph, 2017). The range of coarse particle (D > 1.0 μm) and number concentrations monitored by an optical particle counter (OPC KC-01E, RION, Japan) at the NOTO Ground-based Research Observatory (NOTOGRO; 37.45˚N, 137.36˚E) (Iwamoto et al., 2016) during the sampling periods was



significantly higher than the background concentrations ($0.31 \pm 0.12$ particles/cm$^3$) during the 2016 spring season. Although the observed concentrations in February were not as high as that expected during major Asian dust events ($3.27 \pm 1.80$ particles/cm$^3$), the sporadic nature of the increasing particle concentration and air-mass transport patterns suggested that the samples were collected under the influence of a modest Asian dust event. The April sampling date is reported as an Asian

dust event day by the Japanese Meteorological Agency (http://www.data.jma.go.jp/gmd/env/kosahp/kosa_table_2016.html). Indeed, the coarse particle concentration in April ($2.45 \pm 0.09$ particles/cm$^3$) was three times as high as that in the February ($0.85 \pm 0.08$ particles/cm$^3$) sampling period at the sampling location, and five times higher ($2.38 \pm 0.07$ particles/cm$^3$) than that in February ($0.49 \pm 0.04$ particles/cm$^3$) at NOTOGRO.

As discussed above, the mineral dust particles were the dominant group of IN active particles. The higher ice nucleation

active fraction observed for the April sample (5.6E -03) compared to the February sample (3.6E -03) can be related to the stronger influence of Asian dust particles.

Based on the result of individual particles analysis, the dust particles collected at the sampling location were internally mixed with sulfates, nitrates, sea salt, and organics. On the other hand, the ADS did not show signs of mixing with sulfates or nitrates, although they did contain either organic matter or BBC. This finding is in agreement with previous reports on Asian

dusts collected at the desert surface, which are reported to contain organics originating from biomass burning and unsaturated fatty acid derived from plants (Kawamura and Gagosian, 1987; Kawamura et al., 2004). Therefore, our ice nucleation experiment is representative of the Asian dust plume, originally emitted from Chinese and Mongolian deserts, experiencing aging during long-range transport across the urban areas of China and/or the Sea of Japan, where particles became internally mixed with sulfates, nitrates, or sea salt (Zhang et al., 2003).

The results of this study revealed that the influences of dust aging and associated mixing during transport on ice nucleation activity is very complex because, in certain cases, the aging process may act to both promote or inhibit ice nucleation in mixed phase clouds. For example, while the mixing of sulfates and nitrates with clay minerals may not significantly affect the original ice nucleation activity, mixing with sea salt was shown to inhibit the IN activity of the mineral dust particles. Moreover, relatively pure calcite and sulfate particles such as $(NH_4)_2SO_4$ may nucleate ice following mixing with other

components (except for the mixing of calcite with nitrate). These results suggest that, in addition to the original composition and related ice nucleation activities, the aging process in the atmosphere must also be considered when precisely predicting the ice nucleation activity of ambient aerosols.

## 5 Conclusions

Ice nucleation experiments on both standard mineral samples and ambient aerosol particles were performed on an individual

particle basis by the individual droplet freezing method, simulating conditions relevant for mixed phase cloud formation. In addition, the morphology and chemical composition of both IN active and non-active particles were directly measured by three individual particle analysis methods.



Standard feldspar and quartz particles were shown to be more ice nucleation active than atmospheric particles. Among the ambient aerosol particles, alumino-silicate mineral dust and internally mixed Ca-rich and sulfate particles were identified as IN active particle types. Other IN active particles were identified as internal mixtures including sulfates, nitrates, and organics, but commonly contained mineral dust as a major component. These mineral dust inclusions were suggested to be

mixtures of several mineral components (including defects and impurities in crystal structure such as mineral clay) rather than single mineral species.

Dust particles consisting of pure mineral components, associated with high ice nucleation activity (e.g. K-feldspar) in previous laboratory experiments, were extremely rare or non-existent in the atmosphere, even under the influence of Asian dust transport. Therefore, the majority of immersion and condensation modes IN particles in the atmosphere were identified

as clay mineral particles on an individual particle basis. Our result suggests that the freezing temperatures of individual ice nuclei in the atmosphere do not show large variation, and fall in a relatively narrow range that can be represented by the ice nucleation activity of clay minerals. This claim is consistent with those inferred from bulk analysis (Kaufmann et al., 2016).

Moreover, aged sea salt, pure calcite, mixtures of Ca-rich spherical particles with nitrates, and pure sulfate particles were found to be less active as ice nuclei. In particular, internal mixing with sea salt particles during transport over the ocean was

shown to be an important factor inhibiting the ice nucleation activity of mixed counterparts. Cloud processing has been proposed as an efficient mechanism for mixing mineral dust and sea salt particles (Niimura et al., 1998), and such depression in the ice nucleation activity has important implications for subsequent condensation and immersion freezing pathways. Although pure calcite and sulfate particles were identified as an inert particle group, interestingly, their internal mixture (i.e. sulfate + Ca-rich) showed relatively higher ice nucleation activity. This also suggests that atmospheric aging (including

cloud processing) could potentially enhance the originally inert ice nucleation activity of calcite and sulfate particles.

The mineral dust particles were found to contain organic matter already at the emission source. Mixing with organics was found to show only a secondary effect on the ice nucleation activity of the host particles. The possibility remains, however, that the oxidation state of the organics and the presence of BBC on the particle surface may be partly involved in ice nucleation. Therefore, there is a further need to characterize the surface and coating state of atmospheric particles to better

understand the specific factors responsible for ice nucleation in mixed phase clouds. This is because ice nucleation is probably most sensitive to the particle surface or the first few layers below the surface, as ice germs are believed to grow from cracks, crevices, or pores.

In conclusion, this study successfully related immersion and condensation mode ice nucleation activity of actual atmospheric particles to their morphology, chemical composition, and mixing states on an individual particle basis. This was made

possible by the direct and comprehensive particle analysis of individual IN particles. We believe that this method can be used to verify previously proposed aerosol ice nucleation theories that are mostly based on experiments using single components and/or bulk samples.



**Acknowledgement**

We thank Shoji Arai, Tomoyuki Mizukami (Kanazawa University), and Norikatsu Akizawa (Kyoto University) for providing experimental equipment. We gratefully acknowledge the Fourth Laboratory, Forecast Research Department, Meteorological Research Institute for providing the ATD and valuable comments. The authors also acknowledge the NOAA

Air Resources Laboratory (ARL) for provision of the HYSPLIT transport and dispersion model and/or the READY website (http://www.ready.noaa.gov) used in our work. This study was supported by the Japan Society for Promotion of Science (JSPS) Funding Program for Next Generation World-Leading Researchers (# GR045).

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

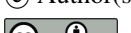



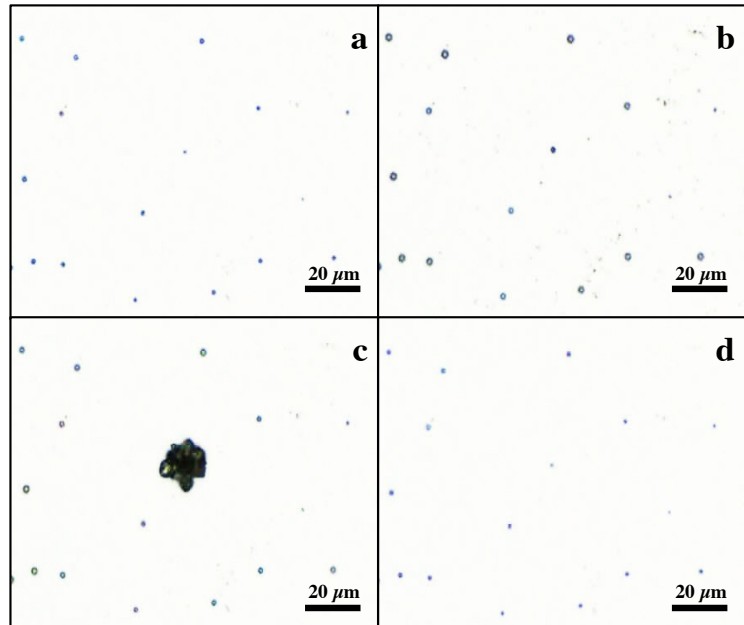

**Figure 1: Optical images of sample particles deposited on Si wafer substrate before the freezing experiment (a), after exposure to water super saturation conditions at -9 ˚C (b), after cooling to -30 ˚C (c), and after evaporation by dry air (d).**





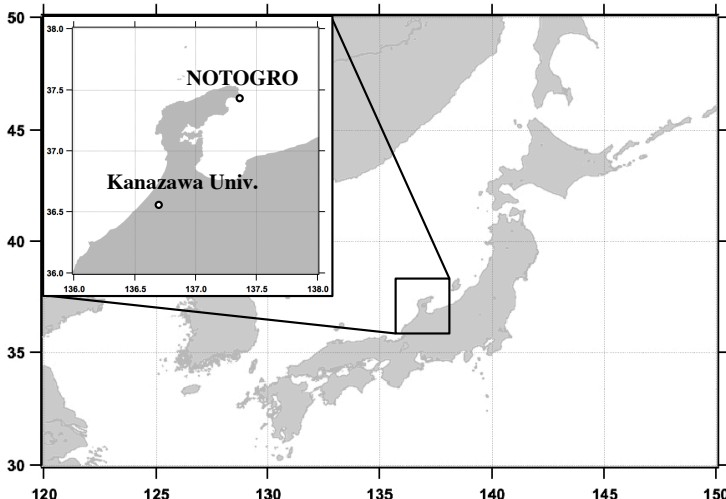

**Figure 2: Sampling location of atmospheric particles (Kanazawa University), and location of the observatory for particle concentration monitoring (NOTO Ground-based Research Observatory: NOTOGRO).**





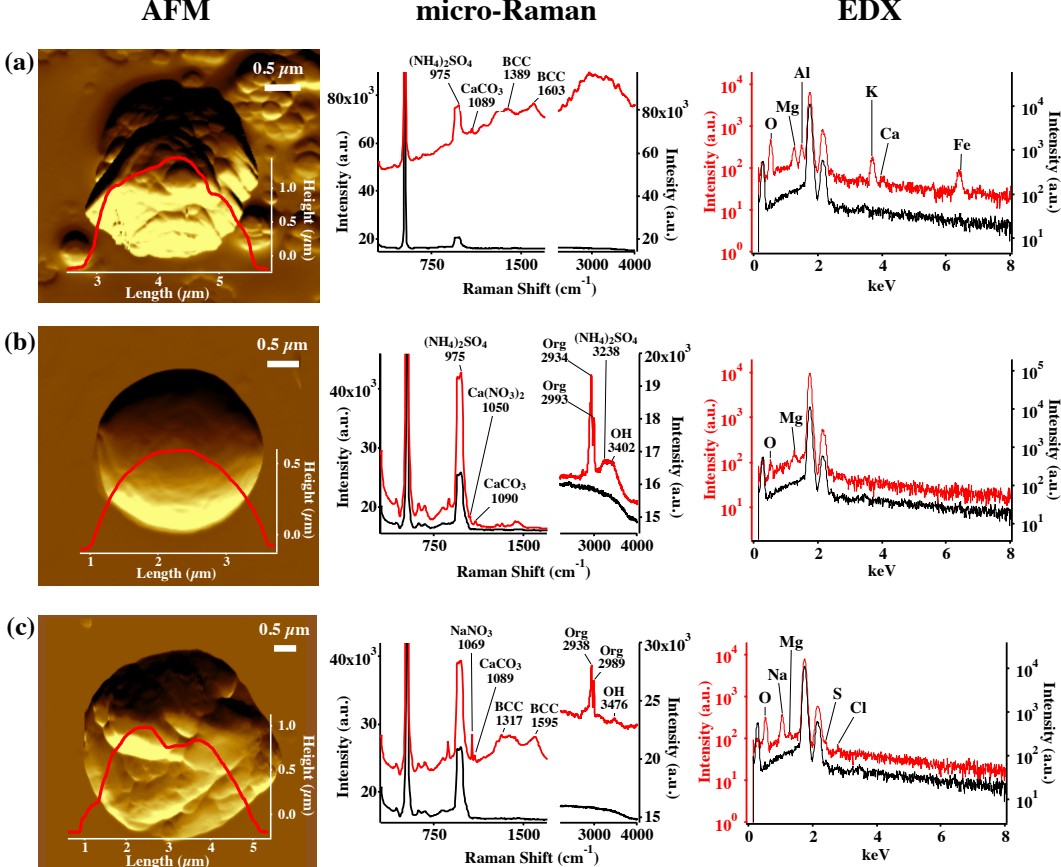

**Figure 3:** AFM topographic images of representative IN active particle (a) and non-active particle (b, c) groups, and their corresponding Raman and EDX spectra. The AFM images were obtained in probe amplitude mode. The inset in the AFM image shows the height of each particle. The red and black curves indicate the spectra of the particles and the substrate background, respectively.



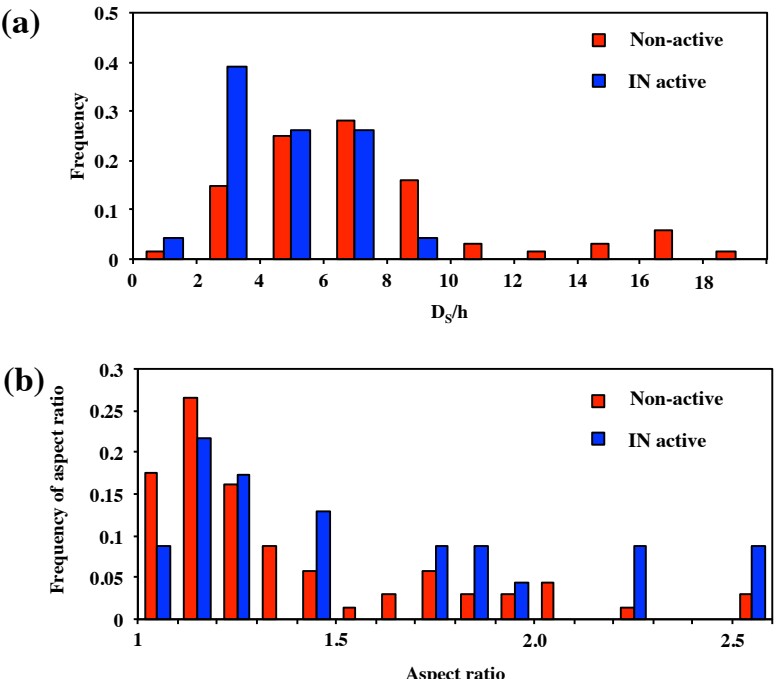

**Figure 4: Frequency distributions of $D_S/h$ (a) and aspect ratio (b), for non-active and IN active particles from AFM observation.**





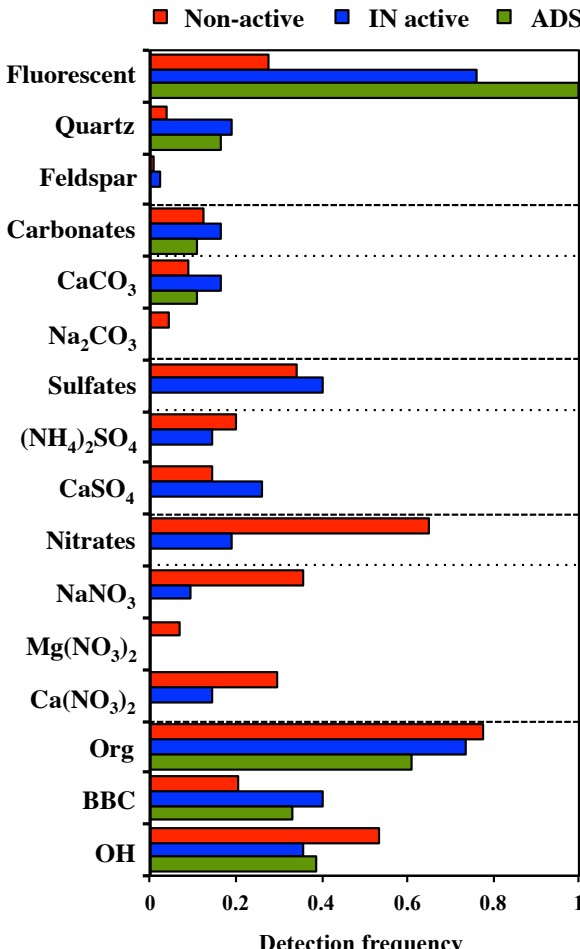

**Figure 5: Summary of the detection frequencies of the assigned components in non-active and IN active particles by micro-Raman analysis. Data from Asian dust source (ADS) particles are shown for comparison. The detection**
5 **frequencies of sulfates, carbonates and nitrates are further classified based on the counter cations (NH₄, Na, Mg, and Ca).**





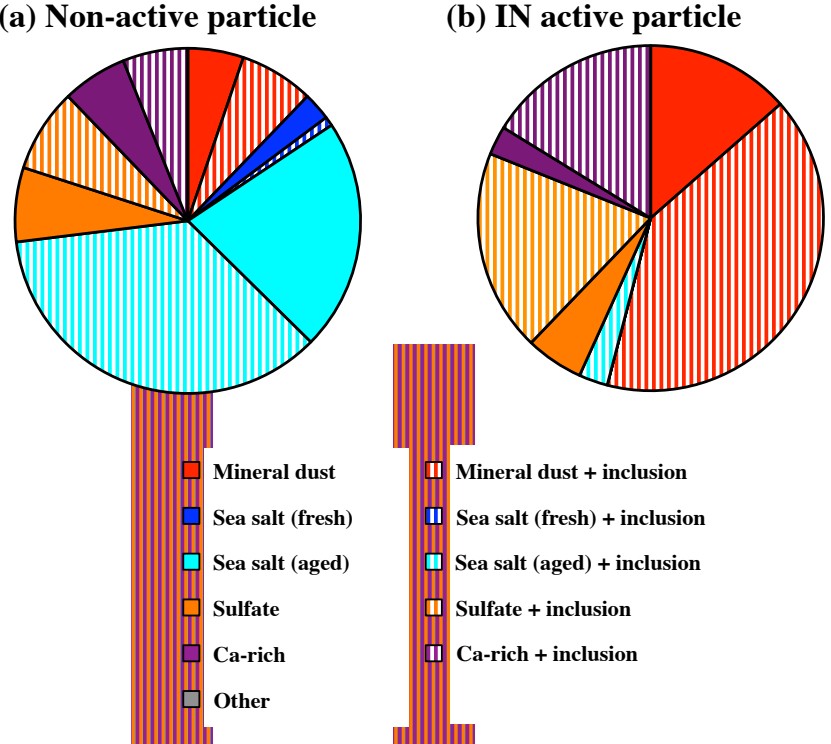

**Figure 6: Relative abundance of the particle groups identified by SEM-EDX for non-active and IN active particles. April and February samples are combined.**





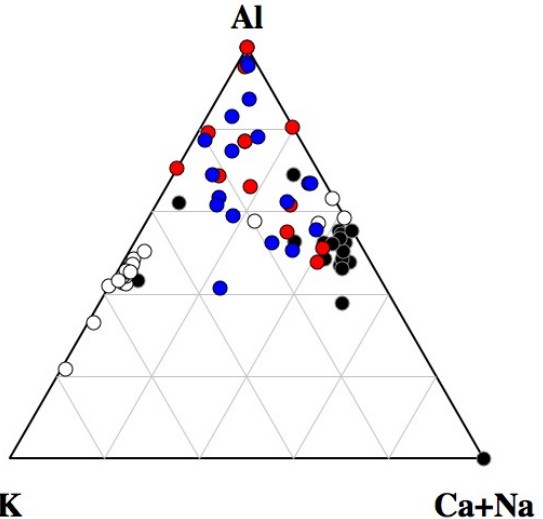

**Figure 7:** Ternary diagram of Al-K-(Ca+Na) components, showing the compositions of mineral dust particles identified by SEM-EDX analysis. Data presented in relative atomic proportions. The circle symbols indicate non-active particles (red) and IN active particles (blue). Particles of Na-feldspar (black) and K-feldspar (white) are also shown for comparison.



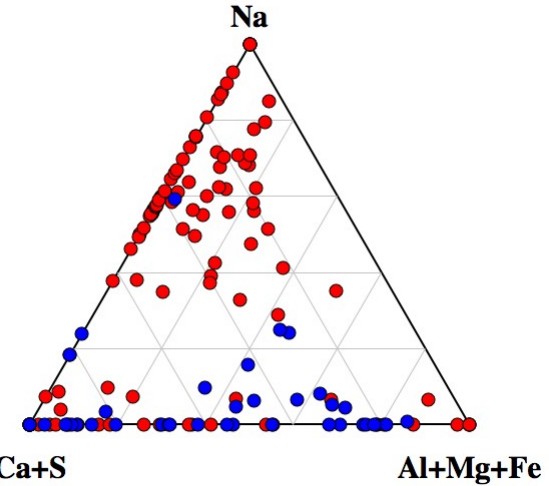

**Figure 8:** Ternary diagram of Na-(Ca+S)-(Al+Mg+Fe) components, showing the compositions of all particles analyzed by SEM-EDX analysis. Data presented in relative atomic proportions. The circle symbols indicate the non-active (red) and IN active particles (blue).



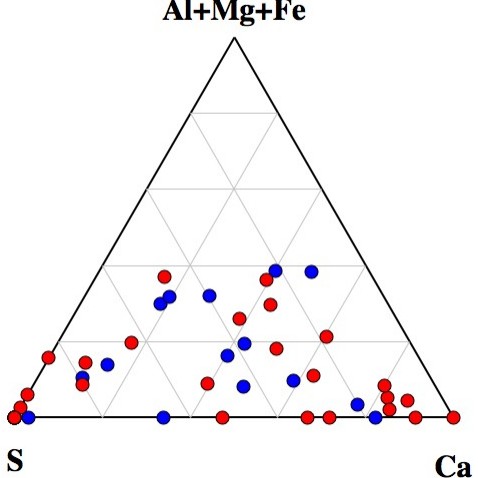

**Figure 9: Ternary diagram of (Al+Mg+Fe)-S-Ca for Ca-rich and sulfate particles analyzed by SEM-EDX analysis. Data presented in relative atomic proportions. The circle symbols indicate the non-active (red) and IN active particles**

5 **(blue).**




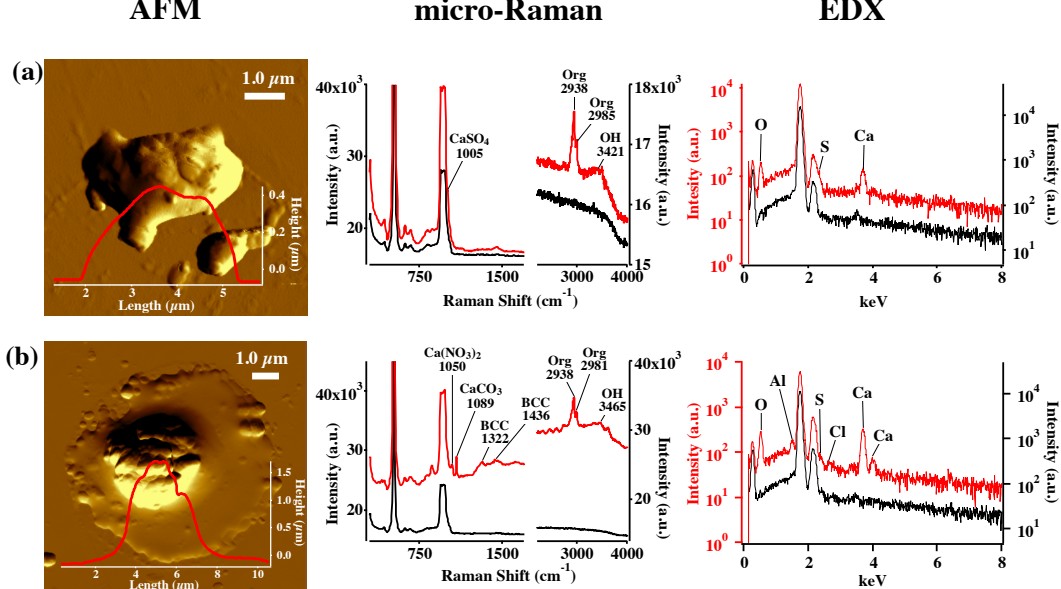

**Figure 10: AFM topographic images of representative Ca-rich particles in IN active (a) and non-active (b) groups, and their corresponding Raman and EDX spectra. The AFM images were obtained in probe amplitude mode. The inset in the AFM image shows the height of each particle. The red and black curves indicate the spectra of the particle and the substrate background, respectively.**





**(a) 28 February 2016**    **(b) 10 April 2016**

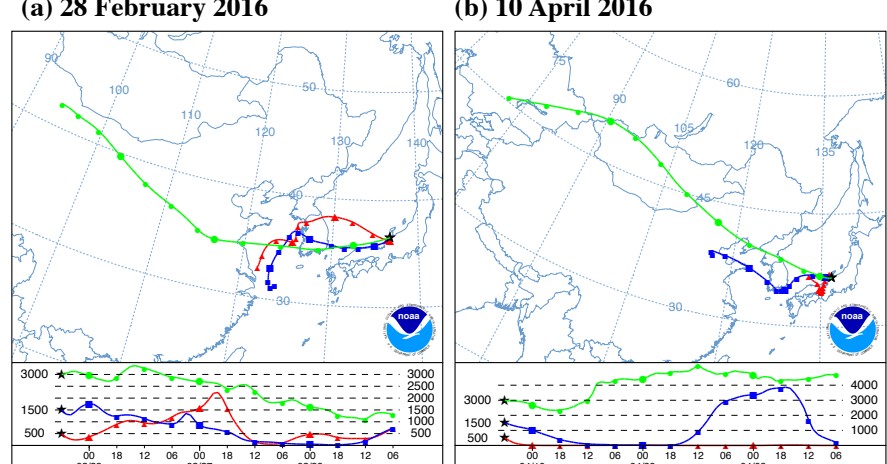

**Figure 11: Backward trajectories of air masses that arrived at 3,000 m, 1,500 m, and 500 m over the sampling location during the sampling periods on the 28 February and 10 April, 2016.**



**Table 1: Peak assignments for Raman spectra obtained in this study.**

| Raman peak assignments | | |
|---|---|---|
| | Raman Shift (cm$^{-1}$) | Literature |
| Feldspar | 485 | Freeman et al., 2008 |
| Quartz | 465 | Laskina et al., 2013 |
| $CaCO_3$ | 1089 | Laskina et al., 2013 |
| $Na_2CO_3$ | 1080 | Hiranuma et al., 2011 |
| $(NH_4)_2SO_4$ | 975 | Tang and Fung, 1989 |
| $CaSO_4$ | 1005 | Hiranuma et al., 2011 |
| $Na_2SO_4$ | 990 | Tang and Fung, 1989 |
| $NaNO_3$ | 1069 | Tang and Fung, 1989 |
| $Mg(NO_3)_2 \cdot 6H_2O$ | 1059 | Tang and Fung, 1989 |
| $Ca(NO_3)_2 \cdot 4H_2O$ | 1050 | Tang and Fung, 1989 |
| C-H vibration (Organics) | 2800 - 3100 | Baustian et al., 2012 |
| Black and brown carbon | 1200 - 1700 | Ivleva et al., 2007 |
| Bonded OH stretch | 3200 - 3650 | Baustian et al., 2012 |
| Fluorescent | - 4000 | |