# Peer review of "Characterization of individual ice residual particles by the single droplet freezing method: a case study in the Asian dust outflow region"

_Atmospheric Chemistry and Physics, 2017_

## Referee Comment (RC1) · Anonymous Referee #3 · 18 Aug 2017

The authors attempted to characterize the physico-chemically relevant ice-nucleating particles with the new off-line approach (i.e., a combination of several off-the-shelf techniques), which could potentially complement the in situ approach of ice residual studies. The topic itself is an important addition to ACP and the atmospheric science community. However, the experimental and analytical methods are unfortunately poorly explained and partly erroneous. Further, the manuscript contains a number of ambiguous statements and over-interpreted results without proper quantitative analyses as well as conclusive performance verifications. I have numerous suggestions for critical revisions. Additional tests of IDFM are necessary, and I do believe that the revision of the manuscript could be time consuming and result in a

significantly different paper.

Major comments and suggestions for additional tests are listed below:

The title is misleading. "Characterization of individual ice nuclei. . ." should read ". . .ice residual. . ." unless the authors could provide the evidence of particular individual particles repeatedly form ice over several cooling-thawing cycles. The properties of the particle may alter during/after ice activation, and the particle should be considered as a residual after thawing.

In the introduction section, the authors need to logically address why it is particularly important to study the aerosol mixing state to improve our understanding of atmospheric ice nucleation in mixed-phase clouds. What is special about the mixing state on ice nucleation as compared to other general properties, such as size and bulk composition? The authors may want to do a careful and rigorous literature review, digest the contents in a diplomatic manner and describe your thoughts to the reader along with your own story line.

Why did the authors arbitrarily pick -30 dC as the end cooling T (no explanation given)? Why didn't the authors carry out the T-binned analyses (e.g., up to -20 dC cooling vs. up to -30 dC cooling)? Such capability (seemingly feasible) and approach can resolve the issue, which the authors point out in the manuscript (e.g., P15L15-23).

Because differences in particle composition may be correlated with particle size, it is difficult to determine which characteristic fundamentally drives cloud-nucleating ability. Why didn't the authors conduct the size dependent analysis (i.e., coarse vs. fine) to examine if the size can be a nucleation-triggering factor? It is not appropriate to generalize the results based on all-size population.

The cooling rate of 0.5 K/s seems aggressive and not atmospherically relevant (i.e., ∼1 K/m may simulate a typical convective cloud updraft). Why not 1 K/min just like other numerous cold stage techniques do?

Show the profiles of dew point and T throughout the experiment rather than show snapshot pictures (i.e., Fig. 1).

Quantitatively evaluate the capability of immersion/condensation freezing method relative to other cold stages using the ice nucleation parameters, such as frozen fraction or ice nucleation active surface site density. The onset discussion (P711-18) seems qualitative, speculative and over-simplified. Description of the background contribution (contamination/impurity) should be included. The authors can assess it by putting soluble salts (e.g., NaCl) on the silicon wafer and cooling the cold stage down to the homogeneous freezing T.

Why the homogeneous freezing occurs over 3.5 K (P7L6-8)? It should be abruptly spontaneous in a narrow range of T, if the water saturation condition is well-controlled. The size of droplets might not be a substantial factor for the observed deviation. Technical validations seem necessary.

Fig 1: What particles are they? Regardless, the immersion freezing active fraction at -30 dC is 1/16. Is it comparable to other cold stage techniques? The authors may want to test reference samples (K-feldspar, quartz etc.) and estimate their $n\_s$ for to quantitatively compare to Atkinson et al. (2013, Nature).

Fig. 6: Two more pie charts should be added. Show the composition of total aerosols measured before cooling as well as that of after thawing to eliminate the artifact of cooling-thawing.

Sea salts are not ice nucleation inhibitors (P1L23), it is just not as active as other known INPs (e.g., aluminosilicate).

P2L9: Koop, 2000 (Nature) missing

How were reference sample powders dispersed/aerosolized onto silicon wafers? Some details of single particle techniques (laser intensity of Raman, and its influence on composition detection) are also missing.

[Figure]

Describing the experimental uncertainties in temperature measurement, particle size detection/limit etc. would be beneficial to the reader.

---

## Referee Comment (RC2) · Anonymous Referee #1 · 21 Aug 2017

Review

**Characterization of individual ice nuclei by the single droplet freezing method: a case study in the Asian dust outflow region**

by Ayumi Iwata and Atsushi Matsuki

This study investigates the ice nucleation characteristics of two samples of Asian dust collected from the west coast of Japan. Particles with diameters > 1.1 μm were deposited on a hydrophobic Si waver using an impactor. The ice nucleation ability of the collected particles was investigated in immersion freezing mode in a closed cell mounted onto a cold stage of an optical microscope. The ice nucleating particles (INPs) together with a number of inactive particles were characterized first with AFM and micro-Raman spectroscopy followed by SEM-EDX. For comparison several reference samples were investigated with the same methods. This study presents a new approach to characterize atmospheric INPs on an individual particle basis. It is a welcome complement to other combinations of methods used to characterize atmospheric INPs on a single particle basis. It shows once more how complex the composition of aged aerosol particles in the coarse mode is, rendering a clear classification of particles difficult. Nevertheless, the authors succeeded to identify physical and chemical characteristics that increase or decrease the ice nucleation ability of the particles. I recommend this paper for publication in ACP after revisions. Specifically, the experimental procedure needs to be explained in more detail. For the validation of the method, the reference samples need to be analyzed more quantitatively and compared more thoroughly with available literature. Moreover, more references need to be added to the introduction and the formulation of some sentences needs to be improved (see specific comments below).

Specific comments

Page 1, line 19: what is meant by "slower"? At lower temperature?

Page 2, line 9: reference to Murray et al. (2012) only is somewhat arbitrary. Add more references or "e.g." in front of the reference to the Murray et al.review..

Page 2, line 21 – 22: This sentence needs to be reformulated.

Page 2, line 23: "sea salt and sulfates are often not considered as efficient ice nuclei.": This is not generally true for sulfates. Ammonium sulfate is found to be ice nucleating at lower temperatures as is also stated later in the manuscript by referring to e.g. Abbatt et al. (2006). The sentence here needs to be corrected, e.g. by adding "under mixed phase conditions".

Page 2, lines 30 – 31: This sentence needs to be improved.

Page 3, lines 1 – 3: this statement should be supported by more recent references.

Page 3, lines 3 – 5: this statement should be supported by a reference.

Page 3, line 12: which "above mentioned artifacts" are meant here? Please, be more explicit.

Page 3, line 20: more than just one reference should be given here to support this statement.

Page 3, lines 21 - 22: again, more than one reference should be given to support this statement.

Page 4, lines 25 – 28: Was the dew point kept constant at -6 to -3°C during cooling? Also, the degree of dilution of the particle should be estimated based on the size of the developing droplet. This is important to judge whether a freezing point depression as discussed later on in the manuscript is relevant at all.

Page 5, lines 26 – 27: This sentence needs to be improved.

Page 7, lines 11 – 16: Here, the freezing onset temperatures of the reference samples are compared with literature (Atkinson et al., 2013) and it is concluded that they are consistent. However, this comparison does not take the surface area present in the samples into account. The comparison needs to be based on the surface area present in the investigated reference sample of this study multiplied with published ice nucleating active sites (INAS) surface densities e.g. from Peckhaus et al. (2016) or Harrison et al. (2016) for feldspars. Moreover, It should be specified what frozen fraction was taken as the onset condition.

Page 7, line 18: The ATD and ADS samples should be discussed in more detail. How many particles were analyzed? Comparison of freezing temperatures with relevant literature is needed.

Page 7, line 19 – 21: the measurement procedure needs to be explained better. Was the same sample cooled repeatedly to investigate the ice nucleation activity of the 10,188 and 24,145 particles? If yes, what were the heating/drying conditions between the cooling cycles to remove the ice crystals?

Page 7, lines 22 – 23: Ice nucleation active fractions should also be given for the reference samples.

Page 7, line 24: specify "onset temperature" in terms of a frozen fraction.

Pages 9 – 10, Section 3.4: The elements C, N, O and Si could not be identified by SEM-EDX. The influence of this restriction should be discussed more explicitly. It should be analyzed whether based on the Raman spectra, some particles might be dominated by organics. In general, it might be helpful to combine the results of the Raman spectra and the SEM-EDX analysis already in the results section and not only in the discussion section. Also, do the compositions derived from the two techniques support each other or are they in some cases contradictory? At the end of this section, it is stated that the Raman peaks are "not necessarily representative of a major component of the particle". This is certainly true for large samples. However, in case of particles with diameters < 5 µm, the laser spot penetrates the whole sample. This is supported by the presence of Raman peaks from the substrate present in the spectra. Moreover, in the method section, it is stated that Raman spectra were taken "each 750 nm step". This should lead to a full coverage of the particles in the x- and y-directions.

Page 10 – 12, Section 4.1: This section needs to be improved. The assignment of clay minerals is solely based on a fluorescence signal in the Raman spectra. However, fluorescence is also often taken as an indication for the presence of biological aerosol particles (e.g. Twohy et al., 2016). How can a biological origin of the fluorescence signal be excluded?

Page 11. lines 5 – 6: An Al-K-(Ca+Na) ternary plot is not a typical way to discriminate K-feldspars from Na/Ca-feldspars. For this purpose a K-Na-Ca plot is usually used.

Page 11, lines 6 – 7: Why should clay minerals and mica appear in the middle of an Al-K-(Ca+Na) ternary plot? The clay mineral kaolinite contains only Al but no K nor Ca/Na. It should appear therefore at the Al corner. The clay mineral illite contains Al and K but no Ca/Na. It should therefore appear on the Al-K line. The same with the common mica muscovite: it contains Al and K but no Ca/Na.

Page 11, line 8: How was the presence of mica inferred?

Page 11, lines 17 – 19: Only one study is cited although the sentence starts with "Previous studies". Either give reference to at least one more study or change to "A previous study".

Page 11, line 23: improve formulation, e.g. reformulate: "most IN particles active above -30°C."

Page 11, lines 23 – 26: The discussion of the origin of the fluorescence signal needs to be improved.

Page 12, lines 7 – 8: It should be stated here, that this number has a low bias because contributions from organics are missed in SEM-EDX. An estimate of the contributions of organics based on the Raman spectra might be added.

Page 12,. line 23: "at any event" seems a too strong conclusion considering that aged sea salt particles + inclusions were also found as a small fraction in the IN active particle fraction.

Page 13, line 30 – 31: "Therefore, the liquefied $Ca(NO_3)_2$ coating is expected to show strong molar depression of freezing point, which could explain their weak IN ability". This argumentation is only valid in the case of concentrated solutions. Its validity in the present case depends on the degree of dilution of the soluble material in the droplet. The dilution should be estimated to test the validity of this argumentation.

Page 14. Line 31: Can the OH peaks be associated with either alcohol or carboxy functionalities?

Page 15, lines 25 – 26: improve this sentence. You might split it: "…mainland Japan. Further upstream…"

Page 16, line 2: why exactly 3.27±1.80 particles/cm$^3$? Please explain.

Page 16, line 8: how far away is NOTOGRO from the sampling location?

Page 16, lines 20 – 27: this paragraph reads like a conclusion and might be deleted here and merged with the conclusion section.

Page 17, line 5: the text in the bracket needs to be formulated better. Is it meant that mineral components such as clay minerals have defects in their crystal structure and contain impurities?

Page 28, Fig. 3 and page, 35, Fig. 10: the line along which the transect of the particle was scanned should be indicated. The length bar of 0.5 μm given on top of the AFM images does not seem to correspond with the length axis given at the bottom in some images. Please check.

Page 30, Fig. 5: what sulfates are meant here if not ammonium sulfate or calcium sulfate?

Technical comments:

Page 3, line 9: "residues" instead of "residue".

Page 8, line 31: Sobanska et al. (2012) is missing from the reference list.

Page 14, line 4: "nucleus" instead of "nuclei"

Page 26: figure caption: "supersaturation" instead of "super saturation".

Page 27, Figure 2: the axis numbers are too small to be readable.

Page 28, Fig. 3 and page, 35, Fig. 10: The label "BCC" in the Raman spectra should read "BBC".

References:

Harrison, A. D., Whale, T. F., Carpenter, M. A., Holden, M. A., Neve, L., O'Sullivan, D., Vergara Temprado, J., and Murray, B. J.: Not all feldspars are equal: a survey of ice nucleating properties across the feldspar group of minerals, Atmos. Chem. Phys., 16, 10927–10940, doi:10.5194/acp-16-10927-2016, 2016.

Peckhaus, A., Kiselev, A., Hiron, T., Ebert, M., and Leisner, T.: A comparative study of K-rich and Na/Ca-rich feldspar ice-nucleating particles in a nanoliter droplet freezing assay, Atmos. Chem. Phys., 16, 11477–11496, doi:10.5194/acp-16-11477-2016, 2016.

Twohy, C. H., McMeeking, G. R., DeMott, P. J., McCluskey, C. S., Hill, T. C. J., Burrows, S. M., Kulkarni, G. R., Tanarhte, M., Kafle, D. N., and Toohey, D. W.: Abundance of fluorescent biological aerosol particles at temperatures conducive to the formation of mixed-phase and cirrus clouds, Atmos. Chem. Phys., 16, 8205–8225, doi:10.5194/acp-16-8205-2016, 2016.

---

## Referee Comment (RC3) · Anonymous Referee #2 · 22 Aug 2017

The manuscript "Characterization of individual ice nuclei by the single droplet freezing method: a case study in the Asian dust outflow region" by Ayumi Iwata and Atsushi Matsuki describes an application of a droplet freezing apparatus to the characterization of the ice nucleating (IN) particles sampled during two Asian dust events. The authors report on the application of several microanalysis techniques (AFM, Micro-Raman, SEM, EDX) to reveal the chemical composition and morphology of the most IN active particle in a subset of a sample, identified by the droplet freezing technique.

A combination of the single-particle microanalysis techniques with droplet freezing assay methods is a very promising development in the field of the ice nucleating particle

research. An important advantage of the method described in the manuscript is the realization of a "single IN particle per water droplet" approach, which greatly reduces uncertainty associated with preparation of suspensions, characterization of particle size and surface distribution, and variability of particle properties across the droplet population, which are the common issues in a conventional cold stage experiment. However, before starting characterizing the most active ice nucleating particles with all the modern and expensive single particle techniques, one has to be completely sure that the particles identified as the most IN active are really such particles. To be honest, I am not convinced that this is the case. I would nevertheless support the publication of the manuscript provided that authors carefully address the multiple critical issues listed below. This might require a tedious re-evaluation of the results or even new experiments because some issues cannot be resolved without repeating the experiments.

General remarks:

1. I would like to draw the author's attention to the both papers of Schrod at al., AMT 2016, and ACP 2017, who essentially describes a similar but much better-characterized setup (FRIDGE) to study the deposition freezing of ice on the atmospheric IN particles collected with an unmanned aircraft over eastern Mediterranean. This group has gone a long way improving their instrument and achieving reliable results, you can learn a lot from them. These papers have to be mentioned and discussed.

2. The method description is incomplete. Many important details are missing, preventing the objective evaluation of method applicability. The only reference supposedly describing the setup (Akizawa et al., 2016), refers to an application of a Linkam stage for mineralogical samples. I wonder how is that related to the IN study discussed in the manuscript. Peckhaus et al., ACP 2016, also used a Linkam stage for a cold stage experiments.

3. The method described in the manuscript allows identifying the first particle within the

field of view that initiated the freezing. This particle is then labeled the "ice nucleating" particle, and the others as "non-nucleating" particles. I wonder what happens with all other particles on the substrate? Do they initiate freezing too? Please explain this clearly in the manuscript.

4. The setup description in the manuscript doesn't have it very clear, but if I understand correctly, the method has a very important limitation. In a closed cell with the humidified gas disconnected from the cell, the droplets are in equilibrium with the water vapor. The moment the first crystal appears, it grows very fast because the vapor is supersaturated with respect to the ice surface. Now that the system contains all three phases of water, droplets start evaporating and some of them would evaporate completely without having a chance to freeze. To prevent complete evaporation, you use the very fast cooling rate (30 K/min), but this reduces the time remaining for the other droplets to freeze to less than 20 s (if the first droplet freeze at -30°C and the homogeneous freezing is over at -40°C). Ice nucleation is a stochastic process described by a rate equation, so that the probability of freezing at given temperature is a function of time spent at this temperature. At such cooling rate, the freezing of "second-best" IN particles can be inhibited by this time-dependent issue and all the droplets will freeze homogeneously concealing the potential freezing activity of the IN particles. A chance of freezing is further reduced by droplets evaporation.

5. In this way, only one ice nucleating particle can be identified per sample. The other, just very slightly less active ice nucleating particles would have no chance to initiate freezing and would be labeled as non-nucleating. This would result in a wrong statistics of the IN vs the non-IN particles, and as a consequence in a biased composition of non-IN active particles. If I am wrong, please show your results in form of "the number of frozen droplets as a function of the substrate temperature". Such curves can be used to retrieve the so-called ice nucleating active site (INAS) densities, that can be better compared with the measurements of other groups.

6. The critical issue discussed in the above comments can be also a consequence of

the Linkam stage design. The standard cell of a Linkam stage has the metal coolant tubing exposed to the environment, making this tubing the coldest spot inside the cell. Water condensed on the tubing during the condensation step must freeze first during the cooling. As explained above, such ice surface would immediately become a strong sink for the water vapor. If the cooling rate is too slow, the liquid droplets condensed on the residual particles would evaporate before having a chance to freeze. I suspect that this is the reason for using a cooling rate of 30 K/min. If this is the case, it is a serious drawback of the system and has to be clearly stated in the manuscript. For the future work, I would recommend enclosing the substrate with the collected particles into a separate cell, so that just this area of the substrate is exposed to the supersaturated vapor.

7. At the cooling rate this high, a strong temperature gradient across the substrate can arise. The freezing temperature can be biased towards low values. Was temperature measured on the surface of a substrate or taken from the Linkam stage internal measurement? How was the temperature of the freezing onset determined? Have you been recording the microscope images? If yes, what was the image acquisition rate? Was it synchronized with temperature measurements? Since this is the first time you report the measurements with the new setup, you should convince the reader that the setup is well characterized.

8. What is the relationship between the field of view of the microscope and the sampling area containing droplets condensed on the aerosol particles? What if the first ice crystals are located outside of the view area? In this case, the initial ice crystals would not be detected but the local vapor pressure would be reduced due to the vapor deposition on the growing crystals, leading to evaporation of droplets and inhibition of freezing. Please give a detailed assessment of this effect.

Given the uncertainty of the method of identifying the most active IN particles, I don't see the point of discussing the single particle microanalysis. If possible, give the maximum detail of the cold stage operation and detection techniques. If the measurements

data permit, present your data not just in form of on-set freezing temperature, but in form of temperature dependent freezing curves. Otherwise, the measurements have to be repeated with a lower or variable cooling rate, and the results analyzed as discussed above.

Schrod, J., Danielczok, A., Weber, D., Ebert, M., Thomson, E. S., and Bingemer, H. G.: Re-evaluating the Frankfurt isothermal static diffusion chamber for ice nucleation, Atmos. Meas. Tech., 9, 1313-1324, doi:10.5194/amt-9-1313-2016, 2016. http://www.atmos-meas-tech.net/9/1313/2016/amt-9-1313-2016.pdf;

Schrod, J., Weber, D., Drücke, J., Keleshis, C., Pikridas, M., Ebert, M., Cvetković, B., Nickovic, S., Marinou, E., Baars, H., Ansmann, A., Vrekoussis, M., Mihalopoulos, N., Sciare, J., Curtius, J., and Bingemer, H. G.: Ice nucleating particles over the Eastern Mediterranean measured by unmanned aircraft systems, Atmos. Chem. Phys., 17, 4817-4835, https://doi.org/10.5194/acp-17-4817-2017, 2017.

Peckhaus, A., Kiselev, A., Hiron, T., Ebert, M., and Leisner, T.: A comparative study of K-rich and Na/Ca-rich feldspar ice-nucleating particles in a nanoliter droplet freezing assay, Atmos. Chem. Phys., 16, 11477-11496, https://doi.org/10.5194/acp-16-11477-2016, 2016.

---

## Author Comment (AC1) · 9 Nov 2017

Anonymous Referee #3 RC1

The authors attempted to characterize the physico-chemical properties of atmospherically relevant ice-nucleating particles with the new off-line approach (i.e., a combination of several off-the-shelf techniques), which could potentially complement the in situ approach of ice residual studies. The topic itself is an important addition to ACP and the atmospheric science community. However, the experimental and analytical methods are unfortunately poorly explained and partly erroneous. Further, the manuscript contains a number of ambiguous statements and over-interpreted results without proper quantitative analyses as well as conclusive performance verifications. I have numerous suggestions for critical revisions. Additional tests of IDFM are necessary, and I do believe that the revision of the manuscript could be time consuming and result in a significantly different paper.

*First of all, we would like to express our gratitude to the reviewer for carefully going through our manuscript and providing highly relevant remarks and suggestions for improvements. We took the comments very seriously, and hence made substantial efforts to conduct additional tests on our method (IDFM). We believe that the major concerns raised in the comments are now addressed. Please find our response to each of the comments below (in blue italic).*

The title is misleading. "Characterization of individual ice nuclei: : :" should read ": : :ice residual: : :" unless the authors could provide the evidence of particular individual particles repeatedly form ice over several cooling-thawing cycles. The properties of the particle may alter during/after ice activation, and the particle should be considered as a residual after thawing.

*In a strict sense it is true that it is an ice residual particle that we characterized. We changed the title as suggested and now we use the term "ice residual particle". Now the title appears as:*
***'Characterization of individual ice residual particles by the single droplet freezing method: a case study in the Asian dust outflow region'***

In the introduction section, the authors need to logically address why it is particularly important to study the aerosol mixing state to improve our understanding of atmospheric ice nucleation in mixed-phase clouds. What is special about the mixing state on ice nucleation as compared to other general properties, such as size and bulk composition?

The authors may want to do a careful and rigorous literature review, digest the contents in a diplomatic manner and describe your thoughts to the reader along with your own story line.

*The extremely small fraction of IN particles in the total atmospheric particles precludes us from using bulk aerosol sample for characterizing the IN property. In addition to the various external mixing states*

expected in the ambient aerosols, we believe that the internal mixing commonly found in the atmosphere is equally important. Therefore, the detailed information on the individual particles is indispensable for better understanding the ice crystals formation in the real atmosphere. *This is now explained more clearly in the introduction (Page 3,Lline 20) of the manuscript.*

*'Internal mixing of aerosols commonly takes place during long-range transport in the atmosphere (e. g. Zhang et al., 2003, Sullivan et al., 2007, and Iwasaka et al., 2009). The surface properties of the internally mixed particles following atmospheric processing (reaction, coagulation and aging) can dramatically change from their original properties (Maring et al., 2003; Trochkine et al., 2003). The internal mixing of particles is an important factor that contributes to the complexity of atmospheric aerosol particles. Although efforts have been made to address the effects of internal mixing on the IN activity of aerosols under conditions relevant for the mixed phase cloud formation (Sullivan et al., 2010; Kulkarni et al., 2014; Augustin-Bauditz et al., 2016), the complexity of the ambient aerosol has not yet been fully represented by the laboratory generated aerosols. Therefore, detailed investigation based on the individual particle analysis is necessary to relate the internal mixing state of aerosols in the actual atmosphere and their IN activity.'*

Why did the authors arbitrarily pick -30 dC as the end cooling T (no explanation given)?
Why didn't the authors carry out the T-binned analyses (e.g., up to -20 dC cooling vs. up to -30 dC cooling)? Such capability (seemingly feasible) and approach can resolve the issue, which the authors point out in the manuscript (e.g., P15L15-23).

*We observed that the atmospheric particles started to form ice crystals only below -25 °C. The reason why we did not go below -30 °C is that it was difficult to keep track on the Non-IN and IN particles due primarily to the fact that initially formed ice crystals were growing too big and the impact on the surrounding particles became too obvious (scavenging and evaporation of droplets around the rapidly growing ice crystals). Also, if we start dividing the already few number of total IN particles into different temperature bins, it further degrades the counting statistics and makes the comparison of particle types more difficult. Therefore, we had to give up discussing on the problems that require the statistical comparisons by the T - bin analysis in the narrow temperature range (e.g. "P15L15-23" which need to clarify the detailed freezing temperature difference among various organics in atmospheric particles).*

Because differences in particle composition may be correlated with particle size, it is difficult to determine which characteristic fundamentally drives cloud-nucleating ability. Why didn't the authors

conduct the size dependent analysis (i.e., coarse vs. fine) to examine if the size can be a nucleation-triggering factor? It is not appropriate to generalize the results based on all-size population.

*We agree on the importance of size dependent analysis. However, the number of IN in the atmosphere is generally extremely small, and as a result the total number of particles we could identify as ice residues became rather limited. If we further group and divide the already few ice residues into different size groups, it becomes even more difficult to compare the ice-nucleating ability among the particles with diverse mixing states and chemical compositions. All of the particles analyzed in this study are already larger than 1.1µm due to technical limitations, and the largest size was 5.47 µm, thus our result is representative only for this rather narrow range of the coarse particles, and it is now explained more clearly in section 3.1 as follows:*

*'With respect to the particle size detection/limit, the impactor already size segregates particles and limit the test particles in the super-micron range. The diameter of the collected atmospheric particles whose ice crystal formation could be monitored ranged between 1.16 and 5.47 µm through the identification by the optical microscope. Meanwhile, the laser spot size (i.e. spatial resolution) of micro Raman spectroscopy approaches the diffraction limit of approximately 1 µm in diameter. All in all, the size of IN active particles that can be analyzed by this method is limited to super micron particles.'*

The cooling rate of 0.5 K/s seems aggressive and not atmospherically relevant (i.e., 1 K/m may simulate a typical convective cloud updraft). Why not 1 K/min just like other numerous cold stage techniques do?

*It is a very important point. We admit that our cooling rate is considerably faster when compared with the typical cooling rate in the convective cloud updraft. Part of the reason we selected the higher cooling rate is to minimize the chance of scavenging and drying nearby droplets because if we select a slower cooling rate, the resulting ice crystal size became too large. To avoid this from happening and facilitate precise isolation of the ice-nucleating particles, we had to choose the faster rate. However, we did confirm that pure water droplets do not freeze until the stage reaches down to -36.5 degrees even at this cooling rate, which coincides rather well with the results by several previous laboratory experiments (Pruppacher and Klett, 1997;Koop et al., 2000; Murray et al., 2010; Murray et al., 2012) and the observations of deep convective clouds (Rosenfeld and Woodley, 2000). The reason for selecting this cooling rate is now explained in section 3.1 as follows:*

*'The advantage of IDFM is that we can keep track and be sure which of the collected single particle was actually nucleating ice. Thus the IN active particles identified by IDFM can be studied in detail by various particle analysis techniques. Comparing IDFM with the other methods such as FRIDGE used*

*primarily to measure ice nuclei concentration (Ardon-Dryer and Levin, 2014; Schrod at al., 2016; Schrod at al., 2017), we have to compromise the accuracy and quantitative evaluation of IN activity since the evaporation of droplets around frozen particles by the Bergeron-Findeisen effect can affect the activated fraction. Therefore, by selecting the higher cooling rate and -30 ˚C as the end cooling temperature, we minimized the evaporation and the scavenging of the droplets around the rapidly growing ice crystals in the experiments of atmospheric particles. By this way, most of the atmospheric particle (excluding those very close to the ice crystals) were not dried and remained as droplets until temperature reached -30 ˚C. Note however, that the selected cooling rate is considerably faster when compared with the typical cooling rate found in the convective cloud updraft. Also, we cannot fully rule out the possibility that droplets very close to an ice crystal may had been fully evaporated.'*

Show the profiles of dew point and T throughout the experiment rather than show snapshot pictures (i.e., Fig. 1).

*We added the corresponding temperature profile and the time when the images were taken as follows. As described in the manuscript, moist air supply is stopped and the cooling stage is isolated during the cooling process, so the dew point is expected to vary and closely follow the stage temperature while the saturated vapor continued to condense onto the activated droplets.*

[Figure]

*Figure 1: Optical images of sample particles deposited on Si wafer substrate before the freezing experiment (a, b), after exposure to water super saturation conditions at -9 ˚C (c), after cooling to -30 ˚C (d), and after sublimation and evaporation by dry air (e). The inset graph shows the stage temperature and the dew point of the wet air introduced into the cell before exposing the stage to water super saturation.*

Quantitatively evaluate the capability of immersion/condensation freezing method relative to other cold stages using the ice nucleation parameters, such as frozen fraction or ice nucleation active surface site density. The onset discussion (P711-18) seems qualitative, speculative and over-simplified.

*We have to admit that our technique is not a perfect method that can resolve all technical issues encountered in the ice nucleation experiments. The biggest advantage of course is that we can keep track and be sure which single particle was actually nucleating ice. In return, we have to sacrifice the accuracy and quantitative evaluation of freezing temperatures based on INAS in relation to the evaporation of droplets in the vicinity of the frozen particles (Bergeron-Findeisen effect), as well as due to the limited number and size (d>1.1µm) of the particles that can be analyzed.*

*The number of ice crystals formed are compromised especially at lower temperatures as the number and size of ice crystals increase and more droplets in the vicinity of the growing ice are subjected to evaporation. When comparing our results with those found in literature, we must also note that strict comparison is difficult because of the uncertainties and differences related to the size distribution and composition of dust particles used in different experiments. Assumptions made to calculate the surface area can also be a source of bias. The K-feldspar used in this study as a reference mineral dust was purchased from the same supplier, but the lot number is different (from different source rock) from the K-feldspar used in Atkinson et al (2013). Therefore, composition of K-feldspar used in the current study and that used in Atkinson et al. (2013) are not exactly the same. Nevertheless, we added the ice nucleation active sites (INAS) of the 4 standard mineral particles measured by the current IDFM (Fig. S-2).*

*The ice nucleation active sites (INAS) obtained in this study agreed within an order of magnitude difference compared to those found in the literature (despite many different experimental conditions). Further, we would like to emphasize that the relative order of the onset temperatures found for different particles is always consistent (i.e. K-feldspar > Na-feldspar > quartz > kaolinite >> pure water) and even the range of the freezing temperatures are not far off from the reported values.*

*It was probably not clearly stated in the manuscript, so the following change was made with regard to the onset temperatures:*

*'Heterogeneous ice nucleation observed in all standard mineral samples tested in this study (K-feldspar, Na-feldspar, quartz, kaolinite) consistently occurred at higher temperatures than the homogeneous freezing temperature.* The reference mineral samples were milled to fine grains before being collected on Si wafer substrate by an impactor. Three set of samples were made for each reference mineral to ensure large enough observation area for the IDFM. *The total number of the particles monitored during the ice*

*nucleation experiment by IDFM was 4,509, 2,271, 4,759, and 1,435 particles, respectively. In this study, the freezing onset temperature of the sample was defined as the temperature at which the IN active fraction of the total observed particles reached 0.01. As a result, the freezing onset temperatures for K-feldspar, Na-feldspar, quartz, and kaolinite ranged between -22.2 to -24.2 ˚C, -24.7 to -25.7˚C, -24.8 to -26.8 ˚C and -27.2 to -29.2 ˚C, respectively (Fig. S-1). Therefore, the ice nucleation activity of K-feldspar was the highest and that of kaolinite was the lowest. The order and the range of observed onset temperatures for these minerals were consistent with the results found in the literature (Atkinson et al., 2013; Murray et al., 2011).'*

[Figure]

*Figure S-1: Activated fractions of the reference mineral dust particles. Results of the atmospheric samples collected in February and April are also shown for comparison.*

[Figure]

*Figure S-2: the ice nucleation active site (INAS) densities for the reference single component mineral dust samples. These INAS densities were calculated from the activated fractions (Fig. S-1) and the averaged sphere equivalent surface areas obtained from the 2D silhouette of each particles in the microscopic image.*

Description of the background contribution (contamination/impurity) should be included. The authors can assess it by putting soluble salts (e.g., NaCl) on the silicon wafer and cooling the cold stage down to the homogeneous freezing T.

*We are aware of the importance of the background contribution (contamination/impurity). As suggested, we conducted additional measurements on the atomized NaCl particles and the activated fractions as a function of temperature are shown in Fig. S-4. At least, the ice nucleation was not observed above -34 ˚C for both NaCl particles and pure water. Therefore, we believe that there is no significant impact from contamination/impurity both during the water vapor condensation and the cooling processes affecting the ice nucleation down to -30 ˚C.*

[Figure]

*Figure S-4: Activated fractions of NaCl and pure water droplets. Three set of samples were tested for both NaCl and water. The number of particles and droplets observed under the microscope is shown as n. The test NaCl particles were aerosolized by atomizing their solutions (0.005g/ml) and collected on the substrate with an impactor. the pure water droplets were also collected by spraying directly onto the substrate..*

Why the homogeneous freezing occurs over 3.5 K (P7L6-8)? It should be abruptly spontaneous in a narrow range of T, if the water saturation condition is well-controlled.

The size of droplets might not be a substantial factor for the observed deviation. Technical validations seem necessary.

*We believe there was an obvious lack of explanation. In the same freezing run, all droplets freeze in a much narrower range and almost spontaneously at the same temperature, like the reviewer points out (Fig.S-4). -36.5 degrees is the highest $T_{0.5}$ (temperature when freezing fraction is 0.5) that was recorded among the multiple freezing experiments, and all the droplets were frozen before the temperature reached -40 degrees. This explanation is added in the manuscript (Page 7, line 6-8) as follows:*

*'Before evaluating the ice nucleation activity of the standard samples by IDFM, we measured the freezing temperature of pure water droplets using the same method, which can be regarded as the onset temperature of homogeneous freezing. As a result, homogeneous ice nucleation was initiated at approximately -36.5 ˚C or below, and all droplets spontaneously froze in a narrow temperature range within the same experimental run. All droplets froze before the temperature reached down to -40 ˚C in all*

*experimental runs. This homogeneous freezing temperature coincides with those reported by several previous laboratory experiments (Pruppacher and Klett, 1997; Murray et al., 2010; Murray et al., 2012) and from observations of deep convective clouds (Rosenfeld and Woodley, 2000).'*

Fig 1: What particles are they? Regardless, the immersion freezing active fraction at -30 dC is 1/16. Is it comparable to other cold stage techniques?

*The particles shown in Fig 1 are atmospheric particles collected in April, 2016. The original image shown in Fig. 1 was for a small area on the substrate which was enlarged for better visibility. Now Fig. 1 is modified (as shown above in response to earlier 9 comment) and the entire observation area is also shown. The actual activated fractions was 5.6E -03 as stated in the manuscript, which was significantly smaller than 1/16.*

The authors may want to test reference samples (K-feldspar, quartz etc.) and estimate their $n_s$ for to quantitatively compare to Atkinson et al. (2013, Nature).

*As shown in response to earlier comments, the INAS obtained in this study agreed within an order of magnitude difference compared to those found in the literature (Fig. S-2).*

Fig. 6: Two more pie charts should be added. Show the composition of total aerosols measured before cooling as well as that of after thawing to eliminate the artifact of cooling-thawing.

*It would be better if we can show the additional two pie charts, however, conducting analysis before the cooling-thawing experiment will be an additional source of artifacts, for example, SEM analysis involves electron irradiation under high vacuum and can potentially damage the sample and evaporate semi-volatile materials. In addition, it is already quite labor intensive and very difficult to repeat three multiple individual particle analyses on the same particles. The idea was to preserve the original particle properties as much as possible until the cooling experiment. We cannot fully rule out the possibility of the impact of cooling-thawing on the particle properties, but it is beyond the scope of our current study, and must be incorporated in the future course of the study.*

Sea salts are not ice nucleation inhibitors (P1L23), it is just not as active as other known INPs (e.g., aluminosilicate).

*We believe there is also a lack of explanation and overstatement here. The results showed that the particles grouped as mineral dust were the most abundant ice residues in our study, with no obvious dependence on the elemental composition (EDX). On the other hand, most of the particles internally*

*mixed with sea salt were not found as ice residues. This internal mixing counterpart included mineral dust. Therefore, (instead of clearly placing them as an inhibitor) we suggest the internal mixing of mineral dust with sea salt may reduce the chance of the mineral dust to act as ice nuclei.*

P2L9: Koop, 2000 (Nature) missing

*Thank you for pointing out the error. We added it in the reference.*

How were reference sample powders dispersed/aerosolized onto silicon wafers?

*We added the following line to be more explicit on the preparation of the standard mineral samples:*

'The reference mineral samples were milled to fine grains before being collected on Si wafer substrate by an impactor. Three set of samples were made for each reference mineral to ensure large enough observation area for the IDFM.'

Some details of single particle techniques (laser intensity of Raman, and its influence on composition detection) are also missing.

*We added following information in the method section:*

*'The Raman spectra of individual particles were obtained using a 532 nm excitation laser with the intensity fixed at 4.906 mW. This laser wavelength was used to detect C-H vibrational mode that appears as peaks or a broad peak in the range 2,800 $cm^{-1}$ and 3,100 $cm^{-1}$, and the laser intensity was chosen to minimize damage to the sample.'*

Describing the experimental uncertainties in temperature measurement, particle size detection/limit etc. would be beneficial to the reader.

*The temperature measured on the surface of a substrate and the Linkam stage internal measurement is calibrated based on the melting point of materials with known properties as described in Akizawa et al., 2016. The temperature difference between the substrate surface and the stage, as well as the temperature gradient is less than 0.3 °C based on the observation of melting pure ice. This explanation is added in the section 2.1 as follows:*

[revised manuscript text omitted]

---

## Author Comment (AC2) · 9 Nov 2017

Referee#1 comment RC2

This study investigates the ice nucleation characteristics of two samples of Asian dust collected from the west coast of Japan. Particles with diameters > 1.1 $\mu$m were deposited on a hydrophobic Si waver using an impactor. The ice nucleation ability of the collected particles was investigated in immersion freezing mode in a closed cell mounted onto a cold stage of an optical microscope. The ice nucleating particles (INPs) together with a number of inactive particles were characterized first with AFM and micro-Raman spectroscopy followed by SEM-EDX. For comparison several reference samples were investigated with the same methods. This study presents a new approach to characterize atmospheric INPs on an individual particle basis. It is a welcome complement to other combinations of methods used to characterize atmospheric INPs on a single particle basis. It shows once more how complex the composition of aged aerosol particles in the coarse mode is, rendering a clear classification of particles difficult. Nevertheless, the authors succeeded to identify physical and chemical characteristics that increase or decrease the ice nucleation ability of the particles. I recommend this paper for publication in ACP after revisions. Specifically, the experimental procedure needs to be explained in more detail. For the validation of the method, the reference samples need to be analyzed more quantitatively and compared more thoroughly with available literature. Moreover, more references need to be added to the introduction and the formulation of some sentences needs to be improved (see specific comments below).

*The authors would like to thank the reviewer for providing positive comments and constructive suggestions by thoroughly going through the manuscript. As suggested by the reviewer, we added more details on the experimental procedures. We have conducted additional and thorough analysis on the reference samples to validate our method, and quantitatively compared the results with the available literature. Please find our response to each of the comments below (in blue italic).*

Page 1, line 19: what is meant by "slower"? At lower temperature?

*We changed the corresponding sentence as follows:*

*'Meanwhile, most of the IN active atmospheric particles formed ice below -28 °C, i.e. at lower temperatures than the standard mineral dust samples of pure components.'*

Page 2, line 9: reference to Murray et al. (2012) only is somewhat arbitrary. Add more references or "e.g." in front of the reference to the Murray et al.review..

*We added Murray et al., (2010) and Rosenfeld and Woodley, (2000) as the references which also reported that the homogeneous nucleation takes place below -37°C.*

Page 2, line 21 – 22: This sentence needs to be reformulated.

*We changed the corresponding sentence as follows and added few more references:*

*'Based on these results, while mineral dust and biological particles are generally regarded as efficient ice nuclei (Morris et al., 2004; Connolly et al., 2009; Niemand et al., 2012), ice nucleation within mixed phase clouds involving soot and organic particles is still not as clearly demonstrated due to the diverse chemical composition and different experimental conditions (DeMott, 1990; Kireeva et al., 2009).'*

Page 2, line 23: "sea salt and sulfates are often not considered as efficient ice nuclei.": This is not generally true for sulfates. Ammonium sulfate is found to be ice nucleating at lower temperatures as is also stated later in the manuscript by referring to e.g. Abbatt et al. (2006). The sentence here needs to be corrected, e.g. by adding "under mixed phase conditions".

*Indeed, we should state more clearly that we are referring specifically to the mixed phase conditions here. We made the following change as suggested.*

*'On the other hand, sea salt and sulfates are often not considered as efficient ice nuclei under mixed phase conditions.'*

Page 2, lines 30 – 31: This sentence needs to be improved.

*We changed the corresponding sentence as:*

*'This enabled direct and detailed investigation on the particles representative of the actual deposition mode ice nuclei in cirrus clouds.'*

Page 3, lines 1 – 3: this statement should be supported by more recent references.

*We added Prenni et al., (2012) and Ardon-Dryer and Levin, (2014) as the references.*

Page 3, lines 3 – 5: this statement should be supported by a reference.

*We added Korolev, (2007) as the reference.*

Page 3, line 12: which "above mentioned artifacts" are meant here? Please, be more explicit.

*We made the following changes to be more explicit:*

*'However, they also reported discrepancies between the results obtained from the three different sampling techniques, and attributed them to potential bias arising from the artifacts such as the possible generation of particles within the instruments and the inherently scarce number of ice nuclei in the atmosphere.'*

Page 3, line 20: more than just one reference should be given here to support this statement.

*We added Zhang et al., (2003), Sullivan et al., (2007), and Iwasaka et al., (2009) as the references.*

Page 3, lines 21 - 22: again, more than one reference should be given to support this statement.

*We added Maring et al., (2003) as the reference.*

Page 4, lines 25 – 28: Was the dew point kept constant at -6 to -3°C during cooling? Also, the degree of dilution of the particle should be estimated based on the size of the developing droplet. This is important to judge whether a freezing point depression as discussed later on in the manuscript is relevant at all.

*The air supply to the chamber is cut during the cooling phase, so the dew point of the supplied air is not actively controlled or monitored during this time. Nonetheless, as the course of the cooling down to -30°C, the remaining water vapor in the chamber is expected to be maintained slightly above the saturation. This is consistent with the fact that the growth in particle size due to condensation can be visually observed.*

*The dry and wet particle diameters before and after the condensational growth are shown in Table S-1. The degree of dilution was estimated based on the growth rate. As a result, the average concentration of the solution droplets of the atmospheric particles were estimated to be approximately 0.074 g/ml.*

*Table S-1: The diameters and volumes of the particles before (dry) and after (wet) the condensational growth. Corresponding concentrations of the sample and test solute particles in the solution droplets are also shown. Here, $D_{dry}$ and $D_{wet}$ denote the circle equivalent diameters obtained from the 2D silhouette of the particles in the microscopic images taken before and after (at approximately -25 ˚C) the cooling experiment, respectively. The number of the particles in each sample observed by the microscope is shown as n. $V_{dry}$ is the sphere equivalent volume calculated from the corresponding $D_{dry}$. $V_{wet}$ was calculated by assuming droplets having contact angle of 110° relative to the substrate. The droplet volumetric growth factor GF was determined by the ratio of $V_{wet}$ relative to $V_{dry.}$ By assuming following densities (sampled particles: 2.00 $g/cm^3$, NaCl: 2.16 $g/cm^3$, $Ca(NO_3)_2$: 2.36 $g/cm^3$), The calculated mass and molar concentrations of the droplet are shown in terms of m and M, respectively. The test solute particles of NaCl and $Ca(NO_3)_2$ were aerosolized by atomizing their solutions (0.005g/ml) and collected on the substrate with an impactor.*

|  | Sampled particles (n = 144) | NaCl (n = 97) | $Ca(NO_3)_2$ (n = 102) |
|---|---|---|---|
| $D_{dry}$ ($\mu$m) | 3.9 ± 0.8 | 4.8 ± 1.4 | 4.0 ± 1.0 |
| $D_{wet}$ ($\mu$m) | 9.2 ± 1.8 | 14.7 ± 4.7 | 12.9 ± 2.5 |
| $V_{dry}$ ($\mu$m$^3$) | 4.4 ± 2.2 | 9.0 ± 7.6 | 4.9 ± 2.2 |
| $V_{wet}$ ($\mu$m$^3$) | 156.9 ± 75.0 | 750.0 ± 677.0 | 429.2 ± 191.0 |
| GF | 43.9 ± 25.0 | 94.3 ± 55.7 | 123.7 ± 101.4 |
| m (g/ml) | 0.074 ± 0.112 | 0.029 ± 0.015 | 0.024 ± 0.009 |
| M (mol/l) | - | 0.49 ± 0.25 | 0.15 ± 0.05 |

Page 5, lines 26 – 27: This sentence needs to be improved.

*We changed the corresponding sentence as follows:*

*'Both the IN active and non-active particles collected in the atmosphere were analyzed on an individual particle basis using a series of microscopic techniques. Firstly, an AFM (CombiScopeᴛᴍ 1000, AIST-NT, Inc., USA) was used to characterize the three-dimensional morphology, followed by micro-Raman spectroscopy to detect surface chemical compounds.'*

Page 7, lines 11 – 16: Here, the freezing onset temperatures of the reference samples are compared with literature (Atkinson et al., 2013) and it is concluded that they are consistent. However, this comparison does not take the surface area present in the samples into account. The comparison needs to be based on the surface area present in the investigated reference sample of this study multiplied with published ice nucleating active sites (INAS) surface densities e.g. from Peckhaus et al. (2016) or Harrison et al. (2016) for feldspars. Moreover, It should be specified what frozen fraction was taken as the onset condition.

*We have to admit that our technique is not a perfect method that can resolve all technical issues encountered in the ice nucleation experiments. The biggest advantage of course is that we can keep track and be sure which single particle was actually nucleating ice. In return, we have to sacrifice the accuracy and quantitative evaluation of freezing temperatures based on INAS in relation to the evaporation of droplets in the vicinity of the frozen particles (Bergeron-Findeisen effect), as well as due to the limited number and size (d>1.1µm) of the particles that can be analyzed.*

*The number of ice crystals formed are compromised especially at lower temperatures as the number and size of ice crystals increase and more droplets in the vicinity of the growing ice are subjected to*

*evaporation. When comparing our results with those found in literature, we must also note that strict comparison is difficult because of the uncertainties and differences related to the size distribution and composition of dust particles used in different experiments. Assumptions made to calculate the surface area can also be a source of bias. The K-feldspar used in this study as a reference mineral dust was purchased from the same supplier, but the lot number is different (i.e. different source rock) from the K-feldspar used in Atkinson et al (2013). Therefore, composition of K-feldspar used in the current study and that used in Atkinson et al. (2013) are not exactly the same. Nevertheless, we added the ice nucleation active sites (INAS) of the 4 standard mineral particles measured by the current IDFM (Fig. S-2).*

*The ice nucleation active sites (INAS) obtained in this study agreed within an order of magnitude difference compared to those found in the literature (despite many different experimental conditions). Further, we would like to emphasize that the relative order of the onset temperatures found for different particles is always consistent (i.e. K-feldspar > Na-feldspar > quartz > kaolinite >> pure water) and even the range of the freezing temperatures are not far off from the reported values.*

*It was probably not clearly stated in the manuscript, so the following change was made with regard to the onset temperatures:*

*'Heterogeneous ice nucleation observed in all standard mineral samples tested in this study (K-feldspar, Na-feldspar, quartz, kaolinite) consistently occurred at higher temperatures than the homogeneous freezing temperature.* The reference mineral samples were milled to fine grains before being collected on Si wafer substrate by an impactor. Three set of samples were made for each reference mineral to ensure large enough observation area for the IDFM. *The total number of the particles monitored during the ice nucleation experiment by IDFM was 4,509, 2,271, 4,759, and 1,435 particles, respectively. In this study, the freezing onset temperature of the sample was defined as the temperature at which the IN active fraction of the total observed particles reached 0.01. As a result, the freezing onset temperatures for K-feldspar, Na-feldspar, quartz, and kaolinite ranged between -22.2 to -24.2 °C, -24.7 to -25.7°C, -24.8 to -26.8 °C and -27.2 to -29.2 °C, respectively (Fig. S-1). Therefore, the ice nucleation activity of K-feldspar was the highest and that of kaolinite was the lowest. The order and the range of observed onset temperatures for these minerals were consistent with the results found in the literature (Atkinson et al., 2013; Marry et al., 2011).'*

[Figure]

*Figure S-1: Activated fractions of the reference mineral dust particles. Results of the atmospheric samples collected in February and April are also shown for comparison.*

[Figure]

*Figure S-2: the ice nucleation active site (INAS) densities for the reference single component mineral dust samples. These INAS densities were calculated from the activated fractions (Fig. S-1) and the averaged sphere equivalent surface areas obtained from the 2D silhouette of individual particles in the microscopic image.*

Page 7, line 18: The ATD and ADS samples should be discussed in more detail. How many particles were analyzed? Comparison of freezing temperatures with relevant literature is needed.

*In response to the above comment, we added the ice nucleation active sites measured for ATD and ADS, and compared with the natural dust sample found in literature (Niemand et al., 2012)*

*We also made the following change in the manuscript:*

*'For comparison, the freezing onset temperatures of the ATD and ADS were -22.1 to -23.7 ˚C and -25.2 to -27.2 ˚C, for 2,019 and 1,354 monitored particles, respectively.'*

[Figure]

*Figure S-3: the ice nucleation active site densities for ATD and ADS.*

Page 7, line 19 – 21: the measurement procedure needs to be explained better. Was the same sample cooled repeatedly to investigate the ice nucleation activity of the 10,188 and 24,145 particles? If yes, what were the heating/drying conditions between the cooling cycles to remove the ice crystals?

*No we did not repeat the freezing and heating/drying cycles though it would be an interesting experiment to show the robustness and reproducibility of the results. To minimize the impact of repeated activation on the particle properties, we limited it to be just one freezing and heating cycle. Besides, it is very labor intensive to go through that many particles for a repeated number of cycles.*

*The heating/drying condition is described in section 2.1 in the manuscript. We added following explanation in the manuscript:*

*'For ambient samples, we first determined the total number of target particles by analyzing the optical images recorded with x50 magnification. As a result, ice nucleation activity of 10,188 and 24,145*

*particles were monitored by the IDFM for the February and April samples, respectively. To avoid too many particles in contact or in proximity to each other in the same field of view, multiple samples were collected during the sampling period on separate silicon wafer substrates to gain a good total number of particles. '*

Page 7, lines 22 – 23: Ice nucleation active fractions should also be given for the reference samples.
*We added those results in Fig. S-1.*

Page 7, line 24: specify "onset temperature" in terms of a frozen fraction.
*We added the following description in section 3.1 as the definition of the onset temperature.*
*'In this study, the freezing onset temperature of the sample was defined as the temperature at which the ice nucleation active fraction of the total observed particles reached 0.01.'*

Pages 9 – 10, Section 3.4: The elements C, N, O and Si could not be identified by SEM-EDX. The influence of this restriction should be discussed more explicitly. It should be analyzed whether based on the Raman spectra, some particles might be dominated by organics. In general, it might be helpful to combine the results of the Raman spectra and the SEM-EDX analysis already in the results section and not only in the discussion section. Also, do the compositions derived from the two techniques support each other or are they in some cases contradictory? At the end of this section, it is stated that the Raman peaks are "not necessarily representative of a major component of the particle". This is certainly true for large samples. However, in case of particles with diameters < 5 μm, the laser spot penetrates the whole sample. This is supported by the presence of Raman peaks from the substrate present in the spectra. Moreover, in the method section, it is stated that Raman spectra were taken "each 750 nm step". This should lead to a full coverage of the particles in the x- and y-directions.

*The restriction of not being able to take into account C, N, O and Si peaks are that we cannot detect organic containing particles by SEM-EDX analysis alone. As a result, if an organic dominant particle is present, it should be identified by the presence of organic related peaks in Raman and absence of other particular elements in the EDX spectra. This restriction however, does not lead to miscounting of mineral dust particles since even a quartz enriched particle can be detected with inclusions of small amount of Al. We added the following description in section 3.4.*

*'Note however, that elements C, N, O, and Si were not taken into account in the EDX semi-quantitative analysis. Therefore, we cannot rule out the possibility that organic dominant particles may be overlooked by EDX analysis alone.'*

*In relation to the results obtained by Raman spectra and EDX analysis, we found that the compositions derived from the two techniques are generally complementary to each other rather than being contradictory. But it is difficult to obtain a 100% match because the Raman peaks depend on the amount, Raman active cross section and peak overlaps of the compounds. And we also cannot determine if the organic matter is the dominant component in a particle or not solely from the obtained Raman peaks. As pointed out, the following statement "not necessarily representative of a major component of the particle" and the related paragraph (Page 10, 1-9) is omitted from the text since it is true that we cannot prove it. We made following change in the manuscript:*

*'Particle classification with SEM-EDX relies on characteristic X-ray signals, which are used to estimate the major elemental composition of a particle. This information can be considered to reflect the bulk elemental distribution within a particle and has been commonly used for particle classification in many previous studies. In contrast, micro-Raman spectroscopy detects slight shifts of wavelength in the scattered light that reflect the vibrations of molecular bonds specific to the compounds contained in a sample. Therefore the Raman spectrum was used rather to complement the bulk particle type classification by the EDX analysis, and to detect the internally mixed compounds.'*

Page 10 – 12, Section 4.1: This section needs to be improved. The assignment of clay minerals is solely based on a fluorescence signal in the Raman spectra. However, fluorescence is also often taken as an indication for the presence of biological aerosol particles (e.g. Twohy et al., 2016). How can a biological origin of the fluorescence signal be excluded?

*Actually, the assignment of clay minerals is not solely based on a fluorescence signal but also taking into account the subsequent SEM-EDX analysis on the same particles. Our conclusion is based on the fact that 87% of the fluorescent particles contained dust related elements such as Mg, Al, Fe.*

*Indeed, we are aware of the biological fluorescence and it is also stated in Page 8, 29-31 'Fluorescence from a particle is typically attributed to a certain group of organics of biological origin, or the intercalated impurities of humic or humic-like substances in clay minerals and amorphous alumino-silicates (Sobanska et al., 2012; Jung et al., 2014).'*

*Possibility remains that part of the fluorescence results from internal mixing of biological matter attached to the mineral dust. To make it more clear, the following explanation is now added in the manuscript:*

*'Subsequent SEM-EDX analysis showed that most of the fluorescent particles (87 %) contained elements that indicate a mineral composition. The fluorescence is also often taken as an indication for the presence of biological aerosols (Twohy et al., 2016), but considering the fact that all the ADS also showed similar*

*fluorescence, and the relatively small abundance of biological particles as compared to mineral dust in general (Huffman et al., 2012), these fluorescence signals are mostly associated with mineral dust (especially those enriched in clay minerals) rather than pure biological particles. Possibility remains however, that a fraction of the fluorescence signal results from biological matters attached to the clay minerals. In any case, both the SEM-EDX and micro-Raman analyses indicated that mineral dust particles act as efficient ice nuclei under conditions relevant for mixed phase cloud formation.'*

Page 11. lines 5 – 6: An Al-K-(Ca+Na) ternary plot is not a typical way to discriminate K-feldspars from Na/Ca-feldspars. For this purpose a K-Na-Ca plot is usually used.

*The ternary plot in Figure 7 is replaced with the K-Na-Ca plot. In any case, atmospheric particles are not gathering at a certain corner but they are rather scattered and show quite mixed characteristics.*

[Figure]

Page 11, lines 6 – 7: Why should clay minerals and mica appear in the middle of an Al-K-(Ca+Na) ternary plot? The clay mineral kaolinite contains only Al but no K nor Ca/Na. It should appear therefore at the Al corner. The clay mineral illite contains Al and K but no Ca/Na. It should therefore appear on the Al-K line. The same with the common mica muscovite: it contains Al and K but no Ca/Na.

*From our years of experience of SEM-EDX analysis on natural Asian dust aerosols, it is very rare to see a single component dust particle. It is commonly the case that several mineral components are internally mixed within a particle so that plots do not usually stick to Al corner or Al-K line.*

Page 11, line 8: How was the presence of mica inferred?

*According to Yabuki et al., (2003), mica appears in the center rather than along the Al-(Ca+Na+K) line of the ternary Al-(Ca+Na+K)-(Fe+Mg) plot. Most of our dust particles (both IN and non-IN active) also fall in the area where plots of mica, feldspars, clay minerals appear. Most particles do not show characteristic elemental composition of specific mineral types, which is another indication that particles are forming complex mineral mixtures.*

Page 11, lines 17 – 19: Only one study is cited although the sentence starts with "Previous studies". Either give reference to at least one more study or change to "A previous study".

*We changed to "A previous study".*

Page 11, line 23: improve formulation, e.g. reformulate: "most IN particles active above -30°C."

*Reformulated accordingly to the suggestion as follows.*

*Furthermore, it was demonstrated that most IN particles active above -30 °C were mineral dust particles composed mainly of clay minerals, with or without minor mixing of other mineral components, that involve fluorescence, most likely as a result of defects and/or impurities (e.g. humic organics) in their crystal structure (Gaft et al., 2005; Jung et al., 2014; Sovanska et al., 2014).*

Page 11, lines 23 – 26: The discussion of the origin of the fluorescence signal needs to be improved.

*The fluorescence signal may come naturally from the crystal structural defect and/or the organic materials contained in the clay mineral particles. As discussed in section 4.4.(page 16, lines 12-16 in the original manuscript), we even found that all the fresh ADS (Asian dust particles believed to be dominated largely by clay minerals) were fluorescent and contained organic matter to some extent. Therefore the fluorescent particles associated with elements of crustal origin (by EDX analysis) were suggested as clay minerals. We made the following changes to be more explicit:*

'Furthermore, we even found that all the fresh ADS particles (believed to be dominated largely by clay minerals) showed the fluorescence. This fluorescence is likely derived from the defects and/or impurities (e.g. humic organics) in their crystal structure (Gaft et al., 2005; Jung et al., 2014; Sovanska et al., 2014). Therefore, the fluorescent particles associated with elements of crustal origin (by EDX analysis) were

regarded as clay minerals. It was also demonstrated that the most IN particles active above -30 °C were dominated by such fluorescence mineral dust particles.'

Page 12, lines 7 – 8: It should be stated here, that this number has a low bias because contributions from organics are missed in SEM-EDX. An estimate of the contributions of organics based on the Raman spectra might be added.

*Following sentence is added to the corresponding section:*

*Note however, that this elemental fraction of sea salt components is based on SEM-EDX and does not reflect potential contribution from organics and therefore must be regarded as the upper limit.*

Page 12,. line 23: "at any event" seems a too strong conclusion considering that aged sea salt particles + inclusions were also found as a small fraction in the IN active particle fraction.

*We changed the corresponding part as follows:*

*Based on the results of this study, we suggest that large and aged sea salt particles internally mixed with sulfates, nitrates, or organics are less likely to nucleate ice, although the possibility remains for ice nucleation by pure sea salt and sea spray organic particles in the atmosphere.*

Page 13, line 30 – 31: "Therefore, the liquefied Ca(NO3)2 coating is expected to show strong molar depression of freezing point, which could explain their weak IN ability". This argumentation is only valid in the case of concentrated solutions. Its validity in the present case depends on the degree of dilution of the soluble material in the droplet. The dilution should be estimated to test the validity of this argumentation.

*As shown in Table S-1, the average mass concentration of the suspension droplets formed by the atmospheric particles was estimated to be 0.074 g / ml. Similarly, the mass concentrations of droplets formed by pure NaCl and $Ca(NO_3)_2$ measured for comparison were 0.029 g / ml and 0.024 g/ ml, respectively. These concentrations are equivalent to 0.49 mol / l and 0.15 mol / l.*

*We conducted additional experiments to compare the freezing temperatures of pure water and NaCl solution and found that the droplets formed by NaCl particles froze consistently at lower temperatures than the pure water droplets by the current method (Fig. S-4). this result suggests that the molar depression of freezing point by NaCl or $Ca (NO_3)_2$ may occur within the range of concentrations expected in the current ice crystal formation experiment, unless there is an inclusion of extremely active ice nuclei. Furthermore, Pruppacher and Neiburger (1963) suggested that the freezing point depression by inorganic salt may be significant with concentrations down to $10^{-3}$ mol / l. Therefore, with the degree of*

*dilution involved in the current experiment, freezing point depression may be effective and act to hinder freezing of droplets activated from NaCl or Ca (NO₃)₂ enriched particles.*

[Figure]

*Figure S-4: Activated fractions of NaCl and pure water droplets. Three set of samples were tested for both NaCl and water. The number of particles and droplets observed under the microscope is shown as n. The test NaCl particles were aerosolized by atomizing their solutions (0.005g/ml) and collected on the substrate with an impactor. The pure water droplets were also collected by spraying directly onto the substrate.*

Page 14. Line 31: Can the OH peaks be associated with either alcohol or carboxy functionalities?

*Detections of both the OH peak (3200 – 3650 cm⁻¹) and the organic peak (2800 – 3100 cm⁻¹) in this study suggest that the detected particles are highly oxidized organic matter containing hydroxyl group (Laskina et al., 2013). Although it is difficult to draw any conclusions, these Raman spectral features are also similar to OH bands observed for glassy oxygenated organic material (Tong et al., 2011; Baustian et al., 2012).*

Page 15, lines 25 – 26: improve this sentence. You might split it: "…mainland Japan. Further upstream…"

*The sentence was now split accordingly to the suggestion as the follows:*

*'The study area, Kanazawa City, is located along the west coast of mainland Japan. Further upstream of the westerly continental outflow are the vast arid regions of inland China and Mongolia. Every spring, frequent dust outbreaks are observed, transporting a massive amount of mineral dust aerosols (Asian*

*dust) across the region and beyond (Iwasaka et al., 2009).'*

Page 16, line 2: why exactly 3.27±1.80 particles/cm3? Please explain.

*This value is basically what we observed as an average concentration during Asian dust events in February 2016 at our monitoring station. This is explained in the manuscript for better clarity as follows:*
*'The average concentration observed during Asian dust events in February 2016 was 3.27 ± 1.80 particles/cm3, so the concentrations observed during February sampling period were not as high as that expected during major Asian dust events '*

Page 16, line 8: how far away is NOTOGRO from the sampling location?

*We added the following information in the manuscript:*
 *(NOTOGRO; 37.45°N, 137.36°E; 116km north east of the sampling location)*

Page 16, lines 20 – 27: this paragraph reads like a conclusion and might be deleted here and merged with the conclusion section.

*The corresponding paragraph is now merged with the conclusion.*

Page 17, line 5: the text in the bracket needs to be formulated better. Is it meant that mineral components such as clay minerals have defects in their crystal structure and contain impurities?

*Yes, the corresponding part is now changed accordingly to the suggestion:*
*These mineral dust particles were suggested to be mixtures of several clay mineral components rather than single mineral species, having defects in their crystal structure and contain impurities.*

Page 28, Fig. 3 and page, 35, Fig. 10: the line along which the transect of the particle was scanned should be indicated. The length bar of 0.5 μm given on top of the AFM images does not seem to correspond with the length axis given at the bottom in some images. Please check.

*We checked and changed Fig.3 and Fig. 10 as follows:*

**AFM**  **micro-Raman**  **EDX**

[Figure]

Figure 3: AFM topographic images of representative IN active particle (a) and non-active particle (b, c) groups, and their corresponding Raman and EDX spectra. The AFM images were obtained in probe amplitude mode. The inset in the AFM image shows the scanned height along the white transect of each particle. The red and black curves indicate the spectra of the particles and the substrate background, respectively.

[Figure]

*Figure 10: AFM topographic images of representative Ca-rich particles in IN active (a) and non-active (b) groups, and their corresponding Raman and EDX spectra. The AFM images were obtained in probe amplitude mode. The inset in the AFM image shows the scanned height along the white transect of each particle. The red and black curves indicate the spectra of the particle and the substrate background, respectively*

Page 30, Fig. 5: what sulfates are meant here if not ammonium sulfate or calcium sulfate?

*This sulfates in Fig. 5 means the detection frequency of either ammonium sulfate or calcium sulfate. We also changed Fig. 5 to for better clarity as the follows:*

[Figure]

*Figure 5: Summary of the detection frequencies of the assigned components in non-active and IN active particles by micro-Raman analysis. Data from Asian dust source (ADS) particles are shown for comparison.*

Page 3, line 9: "residues" instead of "residue".
*Corrected accordingly.*

Page 8, line 31: Sobanska et al. (2012) is missing from the reference list.
*Corrected accordingly.*

Page 14, line 4: "nucleus" instead of "nuclei"
*Corrected accordingly.*

Page 26: figure caption: "supersaturation" instead of "super saturation".

*Corrected accordingly.*

Page 27, Figure 2: the axis numbers are too small to be readable.

*Corrected accordingly as follow.*

[Figure]

Page 28, Fig. 3 and page, 35, Fig. 10: The label "BCC" in the Raman spectra should read "BBC".

*Corrected accordingly.*

References:

[revised manuscript text omitted]

---

## Author Comment (AC3) · 9 Nov 2017

Anonymous Referee #2 RC3

The manuscript "Characterization of individual ice nuclei by the single droplet freezing method: a case study in the Asian dust outflow region" by Ayumi Iwata and Atsushi Matsuki describes an application of a droplet freezing apparatus to the characterization of the ice nucleating (IN) particles sampled during two Asian dust events. The authors report on the application of several microanalysis techniques (AFM, Micro-Raman, SEM, EDX) to reveal the chemical composition and morphology of the most IN active particle in a subset of a sample, identified by the droplet freezing technique. A combination of the single-particle microanalysis techniques with droplet freezing assay methods is a very promising development in the field of the ice nucleating particle research. An important advantage of the method described in the manuscript is the realization of a "single IN particle per water droplet" approach, which greatly reduces uncertainty associated with preparation of suspensions, characterization of particle size and surface distribution, and variability of particle properties across the droplet population, which are the common issues in a conventional cold stage experiment. However, before starting characterizing the most active ice nucleating particles with all the modern and expensive single particle techniques, one has to be completely sure that the particles identified as the most IN active are really such particles. To be honest, I am not convinced that this is the case. I would nevertheless support the publication of the manuscript provided that authors carefully address the multiple critical issues listed below. This might require a tedious re-evaluation of the results or even new experiments because some issues cannot be resolved without repeating the experiments.

*We appreciate all the positive comments and valuable feedbacks the reviewer provided us. We carefully examined and addressed the issues raised by the reviewer, which also involved thorough re-evaluation of the results and new experiments to validate our experimental method. We believe that the additional information will help improve the technical soundness of our paper. Please find our response to each of the comments below (in blue italic).*

1. I would like to draw the author's attention to the both papers of Schrod at al., AMT 2016, and ACP 2017, who essentially describes a similar but much better characterized setup (FRIDGE) to study the deposition freezing of ice on the atmospheric IN particles collected with an unmanned aircraft over eastern Mediterranean. This group has gone a long way improving their instrument and achieving reliable results, you can learn a lot from them. These papers have to be mentioned and discussed.

*Thank you for reminding us of the works using FRIDGE. We have certainly learned a lot form FRIDGE. We added Ardon-Dryer and Levin, (2014), Schrod at al., (2016) and Schrod at al., (2017) as the references which also reported the measurements of the atmospheric ice nuclei using cold stage. Now*

*they are discussed in the context of the limitations and the merits of IDFM, and cited in the result section as follows:*

*'The advantage of IDFM is that we can keep track and be sure which of the collected single particle was actually nucleating ice. Thus the identified IN active particles by IDFM can be studied in detail by various particle analysis techniques. Comparing IDFM with the measurement methods of ice nuclei concentration number using FRIDGE (Ardon-Dryer and Levin, 2014; Schrod at al., 2016; Schrod at al., 2016), we have to sacrifice the accuracy, quantitative evaluation, and cooling rate of freezing temperatures and humidities based on the activated fraction since the evaporation of droplets around frozen particles by the Bergeron-Findeisen effect. By selected the higher cooling rate and -30 ˚C as the end cooling temperature, we also minimized the evaporation and the scavenging of the droplets around the rapidly growing ice crystals in the experiments of atmospheric particles. Then, most of atmospheric particle (excluding the very close droplets of ice crystals) were not completely dried the formed droplets until temperature reached -30 ˚C. Note that, however, the selected cooling rate is considerably faster when compared with the typical cooling rate in the convective cloud updraft.'*

2. The method description is incomplete. Many important details are missing, preventing the objective evaluation of method applicability. The only reference supposedly describing the setup (Akizawa et al., 2016), refers to an application of a Linkam stage for mineralogical samples. I wonder how is that related to the IN study discussed in the manuscript. Peckhaus et al., ACP 2016, also used a Linkam stage for a cold stage experiments.

*We added and changed the introduction and the method section as follows:*

*'Several laboratory studies used similar cold stage to test the ice nucleation activities of various atmospherically relevant standard particles (Fornea et al., 2009; Baustian et al., 2010; Mason et al., 2015; Whale et al., 2015; Knopf et al., 2014), but not enough studies have been made so far to investigate on the immersion-mode ice nucleation (mixed phase cloud) by the individual particles in the actual atmosphere.'*

*'The sample particles were deposited onto a Si wafer substrate with a hydrophobic coating (Glaco, Soft99 Corporation, Japan). Particles were observed for their position, size, and shape under an optical microscope with x50 magnification (Olympus, Japan) as shown in Fig. 1a. Subsequently, the substrate was transferred onto a cold stage in a closed cell (THMSG600, Linkam Scientific Instruments, UK). Since the cold stage used in this study is cooled by liquid nitrogen, the exposed tube through which the liquid nitrogen passes in the cold cell becomes a cold trap which can act as an additional sink for the water vapor. Therefore, in this study, all cooling parts except the cold stage surface were covered by insulating*

*material. The temperature measured at the cold stage was calibrated by the substances of known melting points (Akizawa et al., 2016). Furthermore, we confirmed that the temperature gap between the substrate and the cold stage was consistently smaller than 0.3 ˚C by observing the melting of water. During the ice nucleation experiment by atmospheric particles, the stage temperature and the dew point were recorded every 1 seconds. The temperature measurement and the images were synchronized with the PC internal clock.'*

*'*

3. The method described in the manuscript allows identifying the first particle within the field of view that initiated the freezing. This particle is then labeled the "ice nucleating" particle, and the others as "non-nucleating" particles. I wonder what happens with all other particles on the substrate? Do they initiate freezing too? Please explain this clearly in the manuscript.

*It is a very important point. We do not have uniform distribution of particles on the substrate, because impinging jet of the impactor concentrates particles closer to the nozzle center, and there is a hotspot where we see too many particles that are close to each other. We discard these areas with too many (right in the center) or too sparse particles (far from center), so we do not keep track on what happened to all other particles on the substrate outside the field of view. We do see cases with multiple particles freezing in the same field of view by monitoring the collected particles in a relatively wide area (Fig.1) and it is not necessarily only one particle that we are labeling as "ice nucleating". We added this in the method section as follows:*

*'By comparing the optical images before and after the ice nucleation experiments, the individual particles that formed ice crystals (excluding those coalescing with adjacent droplets or crystals) were identified and regarded as IN active particles. Most of the particles collected on the substrates were monitored under an optical microscope with x5 magnification (Fig. 1a). We did see multiple particles freezing in the same field of view. However, we did not cover all of the collected particles. Therefore we must note that the Non-active particles or IN active particles outside of our field of view are not included in our counts.'*

[Figure]

*Figure 1: Optical images of sample particles deposited on Si wafer substrate before the freezing experiment (a, b), after exposure to water super saturation conditions at -9 ˚C (c), after cooling to -30 ˚C (d), and after sublimation and evaporation by dry air (e). The inset graph shows the stage temperature and the dew point of the wet air introduced into the cell before exposing the stage to water super saturation.*

4. The setup description in the manuscript doesn't have it very clear, but if I understand correctly, the method has a very important limitation. In a closed cell with the humidified gas disconnected from the cell, the droplets are in equilibrium with the water vapor. The moment the first crystal appears, it grows very fast because the vapor is supersaturated with respect to the ice surface. Now that the system contains all three phases of water, droplets start evaporating and some of them would evaporate completely without having a chance to freeze. To prevent complete evaporation, you use the very fast cooling rate (30 K/min), but this reduces the time remaining for the other droplets to freeze to less than 20 s (if the first droplet freeze at -30_C and the homogeneous freezing is over at -40 ˚C). Ice nucleation is a stochastic process described by a rate equation, so that the probability of freezing at given temperature is a function of time spent at this temperature. At such cooling rate, the freezing of "second-best" IN particles can be inhibited by this time-dependent issue and all the droplets will freeze homogeneously concealing the potential freezing activity of the IN particles. A chance of freezing is further reduced by droplets evaporation.

*It is a very important point. It is true that part of the reason we selected the higher cooling rate is to prevent the evaporation of the droplets. Another reason is to restrict the final size of the growing ice crystals so that they do not coagulate with, or dry many of the surrounding droplets. By selected the higher cooling rate and -30 ˚C as the end cooling temperature, we also minimized the evaporation and*

*the scavenging of the droplets around the rapidly growing ice crystals in the experiments of atmospheric particles. Then, most of atmospheric particle (excluding the very close droplets of ice crystals) were not completely dried the formed droplets until temperature reached -30 °C. However, as the reviewer points out, we cannot fully rule out the possibility that droplets very close to an ice crystal may be fully evaporated. Therefore, it is possible that a moderately IN active particle may be miss-interpreted as non-active particles. We admit that the current method still has limitations and space for future improvements with respect to the quantification and accuracy of the IN counting. The fast cooling rate may also cause "second-best" IN to be overlooked in the current method. These limitations are now clearly explained in the section 3.1 (also as a response to the first comment). However, our primary objective here is to identify the properties of the individual ice nucleating particle in the actual atmosphere, by realizing a clear-cut identification of IN active particles. Therefore, we believe being able to identify even the most active fraction (with potential miss counting of the "second best" IN) from the actual atmosphere is still an important step forward towards our understanding of the IN behaviors of atmospheric aerosols, which are often found in complex mixtures (both internally and externally).*

5. In this way, only one ice nucleating particle can be identified per sample. The other, just very slightly less active ice nucleating particles would have no chance to initiate freezing and would be labeled as non-nucleating. This would result in a wrong statistics of the IN vs the non-IN particles, and as a consequence in a biased composition of non- IN active particles. If I am wrong, please show your results in form of "the number of frozen droplets as a function of the substrate temperature". Such curves can be used to retrieve the so-called ice nucleating active site (INAS) densities, that can be better compared with the measurements of other groups.

*We have to admit that our technique is not a perfect method that can resolve all technical issues encountered in the ice nucleation experiments. The biggest advantage of course is that we can keep track and be sure which single particle was actually nucleating ice. In return, we cannot fully rule out the possibility that the "second-best" IN particles may had been overlooked due to the evaporation of droplets in the vicinity of the frozen particles. Therefore, accuracy and the quantitative evaluation of freezing temperatures based on the activated fraction and the INAS could be biased to some extent. The number of ice crystals formed are compromised especially at lower temperatures as the number and size of ice crystals increase and more droplets in the vicinity of the growing ice are subjected to evaporation.*

*In this study, only a small number of particles collected in the atmosphere formed ice crystals in a limited temperature range between -26 °C and - 30 °C. so it is difficult to present the "the number of frozen droplets as a function of the substrate temperature". Instead, we show INAS and activated fraction of*

*standard samples, and the validity of this experimental method is discussed by comparing these result with previous studies.*

*When comparing our results with those found in literature, we must also note that strict comparison is difficult because of the uncertainties and differences related to the size distribution and composition of dust particles used in different experiments. Assumptions made to calculate the surface area can also be a source of bias. The K-feldspar used in this study as a reference mineral dust was purchased from the same supplier, but the lot number is different (from different source rock) from the K-feldspar used in Atkinson et al (2013). Therefore, composition of K-feldspar used in the current study and that used in Atkinson et al. (2013) are not exactly the same. Nevertheless, we added the ice nucleation active sites (INAS) of the 4 standard mineral particles measured by the current IDFM (Fig. S-2).*

*The ice nucleation active sites (INAS) obtained in this study agreed within an order of magnitude difference compared to those found in the literature(despite many different experimental conditions). Further, we would like to emphasize that the relative order of the onset temperatures found for different particles is always consistent (i.e. K-feldspar > Na-feldspar > quartz > kaolinite >> pure water) and even the range of the freezing temperatures are not far off from the reported values.*

*It was probably not clearly stated in the manuscript, so the following change was made with regard to the onset temperatures:*

*'Heterogeneous ice nucleation observed in all standard mineral samples tested in this study (K-feldspar, Na-feldspar, quartz, kaolinite) consistently occurred at higher temperatures than the homogeneous freezing temperature.* To ensure large enough observation area for the IDFM, these reference mineral samples were milled from rock mass or coarse powder state, then were collected on three set of Si wafers by impactor in the clean booth. *The total number of the particles monitored during the ice nucleation experiment by IDFM was 4,509, 2,271, 4,759, and 1,435 particles, respectively. In this study, the freezing onset temperature of the sample was defined as the temperature at which the ice IN active fraction of the total observed particles reached 0.01. As a result, the freezing onset temperatures for K-feldspar, Na-feldspar, quartz, and kaolinite ranged between -22.2 to -24.2 °C, -24.7 to -25.7°C, -24.8 to -26.8 °C and -27.2 to -29.2 °C, respectively (Fig. S-1). Therefore, the ice nucleation activity of K-feldspar was the highest and that of kaolinite was the lowest. The order and the range of observed onset temperatures for these minerals were consistent with the results found in literature (Atkinson et al., 2013; Marry et al., 2011).'*

[Figure]

*Figure S-1: Activated fractions of the reference mineral dust particles. Results of the atmospheric samples collected in February and April are also shown for comparison.*

[Figure]

*Figure S-2: the ice nucleation active site (INAS) densities for the reference single component mineral dust samples. These INAS densities were calculated from the activated fractions (Fig. S-1) and the averaged sphere equivalent surface areas obtained from the 2D silhouette of each particles in the microscopic image.*

6. The critical issue discussed in the above comments can be also a consequence of the Linkam stage design. The standard cell of a Linkam stage has the metal coolant tubing exposed to the environment, making this tubing the coldest spot inside the cell. Water condensed on the tubing during the condensation step must freeze first during the cooling. As explained above, such ice surface would immediately become a strong sink for the water vapor. If the cooling rate is too slow, the liquid droplets condensed on the residual particles would evaporate before having a chance to freeze. I suspect that this is the reason for using a cooling rate of 30 K/min. If this is the case, it is a serious drawback of the system and has to be clearly stated in the manuscript. For the future work, I would recommend enclosing the substrate with the collected particles into a separate cell, so that just this area of the substrate is exposed to the supersaturated vapor.

*We are aware of this issue of the Linkam stage design. Before the experiment, we made a slight modification in the Linkam stage such that the metal coolant tube is covered by the insulating material and made sure that the condensation on this tube does not act as a strong sink for the water vapor. This is better explained in the revised manuscript as follows:*

*'Since the cold stage used in this study is cooled by liquid nitrogen, the tube through which the liquid nitrogen passes in the cold cell becomes a cold trap which can act as an additional sink for the water vapor. Therefore, in this study, all cooling parts except the cold stage surface were covered by insulating material.'*

7. At the cooling rate this high, a strong temperature gradient across the substrate can arise. The freezing temperature can be biased towards low values. Was temperature measured on the surface of a substrate or taken from the Linkam stage internal measurement? How was the temperature of the freezing onset determined? Have you been recording the microscope images? If yes, what was the image acquisition rate? Was it synchronized with temperature measurements? Since this is the first time you report the measurements with the new setup, you should convince the reader that the setup is well characterized.

*The temperature measured on the surface of a substrate and the Linkam stage internal measurement is calibrated based on the melting experiment of some materials as described in Akizawa et al., 2016. The temperature difference between the substrate surface and the stage, as well as the temperature gradient was determined to be less than 0.3 °C based on the observation of melting of pure ice.*

*The microscopic images were recorded with the maximum rate of (5 fps). During the ice nucleation experiment by atmospheric particles, the stage temperature and the dew point were recorded every 1 seconds. The temperature measurement and the images were synchronized with the PC internal clock.*

*This is better explained in the revised manuscript as follows*

*'The sample particles were deposited onto a Si wafer substrate with a hydrophobic coating (Glaco, Soft99 Corporation, Japan). Particles were observed for their position, size, and shape under an optical microscope with x50 magnification (Olympus, Japan) as shown in Fig. 1a. Subsequently, the substrate was transferred onto a cold stage in a closed cell (THMSG600, Linkam Scientific Instruments, UK). Since the cold stage used in this study is cooled by liquid nitrogen, the exposed tube through which the liquid nitrogen passes in the cold cell becomes a cold trap which can act as an additional sink for the water vapor. Therefore, in this study, all cooling parts except the cold stage surface were covered by insulating material. The temperature measured at the cold stage was calibrated by the substances of known melting points (Akizawa et al., 2016). Furthermore, we confirmed that the temperature gap between the substrate and the cold stage was consistently smaller than 0.3 ˚C by observing the melting of pure ice. During the ice nucleation experiment by atmospheric particles, the stage temperature and the dew point were recorded every 1 seconds. The temperature measurement and the images were synchronized with the PC internal clock.'*

8. What is the relationship between the field of view of the microscope and the sampling area containing droplets condensed on the aerosol particles? What if the first ice crystals are located outside of the view area? In this case, the initial ice crystals would not be detected but the local vapor pressure would be reduced due to the vapor deposition on the growing crystals, leading to evaporation of droplets and inhibition of freezing. Please give a detailed assessment of this effect. Given the uncertainty of the method of identifying the most active IN particles, I don't see the point of discussing the single particle microanalysis. If possible, give the maximum detail of the cold stage operation and detection techniques. If the measurements data permit, present your data not just in form of on-set freezing temperature, but in form of temperature dependent freezing curves. Otherwise, the measurements have to be repeated with a lower or variable cooling rate, and the results analyzed as discussed above.

*Monitoring of ice crystal formation by the optical microscope covers most of the sampled atmospheric particles (please refer to our response corresponding to the 3rd comment) By selecting the higher cooling rate, we minimized the evaporation and the scavenging of the droplets around the rapidly growing ice crystals. Then, most of atmospheric particle (excluding the very close droplets of ice crystals) were not completely dried the formed droplets until temperature reached -30 ˚C in this study. Therefore we believe that suppression of ice crystal formation by local vapor pressure reduction has been minimized at least down to -30 ˚C. However, as shown in temperature dependent freezing curve for the standard dust samples in Fig. S-1, the quantification of the activated fraction especially at lower temperatures is*

*compromised as a result of the drying of droplets by growing crystals. In return, this method enables direct identification of individual particles with high IN activity within the ambient samples.*

*We also added the following explanations of the both limitation for detection of particle and quantitative for ice nucleation in IDFM in the section 3.1 as follows:*

*''The advantage of IDFM is that we can keep track and be sure which of the collected single particle was actually nucleating ice. Thus the identified IN active particles by IDFM can be studied in detail by various particle analysis techniques. Comparing IDFM with the measurement methods of ice nuclei concentration number using FRIDGE (Ardon-Dryer and Levin, 2014; Schrod at al., 2016; Schrod at al., 2016), we have to sacrifice the accuracy, quantitative evaluation, and cooling rate of freezing temperatures and humidities based on the activated fraction since the evaporation of droplets around frozen particles by the Bergeron-Findeisen effect. By selected the higher cooling rate and -30 ˚C as the end cooling temperature, we also minimized the evaporation and the scavenging of the droplets around the rapidly growing ice crystals in the experiments of atmospheric particles. Then, most of atmospheric particle (excluding the very close droplets of ice crystals) were not completely dried the formed droplets until temperature reached -30 ˚C. Note that, however, the selected cooling rate is considerably faster when compared with the typical cooling rate in the convective cloud updraft. Also, we cannot fully rule out the possibility that droplets very close to an ice crystal may be fully evaporated.*

*With respect to the particle size detection/limit, further, the impactor already size segregate particles and limit the test particles in the super-micron range. The diameter of the collected atmospheric particles whose ice crystal formation could be monitored was ranged between 1.16 and 5.47 µm in the identification by the optical microscope. Meanwhile, the laser spot size (i.e. spatial resolution) of micro Raman spectroscopy approaches the diffraction limit of approximately 1 µm in diameter. All in all, the size of IN active particles that can be analyzed by this method is limited to super micron particles.'*

References

Akizawa, N., Tamaru, A., Fukushi, K., Yamamoto, J., Mizukami, T., Python, M., and Arai, S.: High-temperature hydrothermal activities around suboceanic Moho: An example from diopsidite and anorthosite in Wadi Fizh, Oman ophiolite, Lithos, 263, 66-87, 2016.

Ardon-Dryer, K., and Levin, Z.: Ground-based measurements of immersion freezing in the eastern Mediterranean, Atmos. Chem. Phys., 14, 5217–5231, doi:10.5194/acp-14-5217-2014, 2014.

Atkinson, J.D., Murray, B.J., Woodhouse, M.T., Whale, T.F., Baustian, K.J., Carslaw, K.S., Dobbie, S., O'Sullivan, D., and Malkin, T.L.: The importance of feldspar for ice nucleation by mineral dust in mixed-phase clouds, Nature, 498, 355-358, doi: 10.1038/nature12278, 2013.

Baustian, K.J., Wise, M.E., and Tolbert, M.A.: Depositional ice nucleation on solid ammonium sulfate and glutaric acid particles, Atmos. Chem. Phys., 10, 2307–2317, 2010.

Fornea, A.P., Brooks, S.D., Dooley, J.B., and Saha, A.: Heterogeneous freezing of ice on atmospheric aerosols containing ash, soot, and soil, J. Geophys. Res., 114, D13201, doi:10.1029/2009JD011958, 2009.

Knopf, D. A., Alpert, P. A., Wang, B., O'Brien, R. E., Kelly, S. T., Laskin, A., Gilles, M. K., and Moffet, R. C.: Microspectroscopic imaging and characterization of individually identified ice nucleating particles from a case field study, J. Geophys. Res. Atmos., 119, JD021866, doi:10.1002/2014JD021866 , 2014.

Mason, R.H., Chou, C., McCluskey, C.S., Levin, E.J.T., Schiller, C.L., Hill, T.C.J., Huffman, J.A., DeMott, P.J., and Bertram, A.K.: The micro-orifice uniform deposit impactor–droplet freezing technique (MOUDI-DFT) for measuring concentrations of ice nucleating particles as a function of size: improvements and initial validation, Atmos. Meas. Tech., 8, 2449–2462, doi: 10.5194/amt-8-2449-2015, 2015.

Murray, B. J., Broadley, S. L., Wilson, T. W., Atkinson, J. D., and Wills, R. H.: Heterogeneous freezing of water droplets containing kaolinite particles, Atmos. Chem. Phys., 11, 4191–4207, doi:10.5194/acp-11-4191-2011, 2011.

Schrod, J., Danielczok, A., Weber, D., Ebert, M., Thomson, E. S., and Bingemer, H. G.: Re-evaluating the Frankfurt isothermal static diffusion chamber for ice nucleation, Atmos. Meas. Tech., 9, 1313–1324, doi:10.5194/amt-9-1313-2016, 2016.

Schrod, J., Weber, D., Drücke, J., Keleshis, C., Pikridas, M., Ebert, M., Cvetkovic, B., Nickovic, S., Marinou, E., Baars, H., Ansmann, A., Vrekoussis, M., Mihalopoulos, N., Sciare, J., Curtius, J., and Bingemer, H. G.: Ice nucleating particles over the Eastern Mediterranean measured by unmanned aircraft systems, Atmos. Chem. Phys., 17, 4817–4835, https://doi.org/10.5194/acp- 17-4817-2017, 2017.

Whale, T.F., Murray, B.J., O'Sullivan, D., Wilson, T.W., Umo, N.S., Baustian, K.J., Atkinson, J.D., Workneh, D.A., and Morris, G.J.: A technique for quantifying heterogeneous ice nucleation in microlitre supercooled water droplets, Atmos. Meas. Tech., 8, 2437–2447, doi: 10.5194/amt-8-2437-2015, 2015.